



# Modeling Atmospheric Ammonia using Agricultural Emissions with Improved Spatial Variability and Temporal Dynamics

Xinrui Ge[1], Martijn Schaap[2], Richard Kranenburg[2], Arjo Segers[2], Gert Jan Reinds[3], Hans Kros[3], Wim de Vries[3]

[1]Environmental Systems Analysis Group, Wageningen University, Wageningen, the Netherlands
[2]Department of Climate, Air and Sustainability, TNO, Utrecht, the Netherlands
[3]Wageningen Environmental Research, Wageningen, the Netherlands

*Correspondence to*: Xinrui Ge (xinrui.ge@wur.nl)

**Abstract.** Ammonia emissions to the atmosphere have increased substantially in Europe since 1960, largely due to the

intensification of agriculture as illustrated by enhanced livestock and use of fertilizers. These associated emissions of reactive nitrogen, particulate matter and acid deposition have contributed to negative societal impacts on human health and terrestrial ecosystems. Due to the limited availability of measurements, emission inventories are used to assess large-scale ammonia emissions from agriculture, creating gridded annual emission maps as well as emission time profiles, both globally and regionally. The modeled emissions are in turn used in chemistry transport models to obtain ammonia concentrations and

depositions. However, current emission inventories usually have relatively low spatial resolutions and coarse categorizations that do not distinguish between fertilization on various crops, grazing, animal housing, and manure storage in its spatial allocation. Furthermore, in assessing the seasonal variation of ammonia emissions, they do not take into account local climatology and agricultural management, which limits the capability to reproduce observed spatial and seasonal variations in the ammonia concentrations.

This paper describes a novel ammonia emission model that quantifies agricultural emissions with improved spatial details and temporal dynamics over the year of 2010, in Germany and Benelux. The spatial allocation was achieved by embedding the agricultural emission model INTEGRATOR into MACC-III, thus accounting for differences in manure and fertilizer application on croplands and grassland, grazing, animal houses and manure storage systems. The more detailed temporal

distribution comes from the integration of the TIMELINES model, which provided predictions of the timing of key agricultural operations including the day of fertilization across Europe. The emission maps and time profiles were imported into LOTOS-EUROS to obtain surface concentrations and total columns for validation. The comparison of surface concentration between modeled output and in-situ measurements illustrated that the updated model has been improved significantly with respect to the temporal variation of ammonia emission, and its performance was more stable and robust. The comparison between

ammonia total columns from remote sensing and simulations showed that there is an overestimation in Southern Germany and underestimation in Northern Germany, which suggested that updating ammonia emission fractions and accounting for manure transport is the direction for further improvement.



## 1 Introduction

Ammonia ($NH_3$) emission to the atmosphere has risen substantially at global scale during the twentieth century following demand for food of a rapidly growing population (Erisman et al., 2008). Increases are especially large in areas with intense agricultural activities, such as Europe, the US and China. The annual European Union emission inventory report 1990-2015

shows that even though ammonia emission of EU-28 countries fell by 23% between 1990 and 2015, Germany, Spain, Sweden and the EU as a whole exceeded their ammonia emission ceilings in 2015 (EEA, 2017). The main source of ammonia emission is agriculture, contributing to more than 90% of the total emissions (Monteny and Hartung, 2007). Ammonia is emitted to the atmosphere during the application of manure and inorganic mineral fertilizers, as well as from animal houses and manure storage systems (Velthof et al., 2012). Additional minor source categories include food processing, biomass burning and fossil

fuel combustion, making up about 4% of the ammonia emissions (Erisman et al., 2008; Galloway et al., 2006; Krupa, 2003). $NH_3$ concentrations are highly variable in space and time because of its short atmospheric residence time as it is effectively removed by dry and wet deposition several hours after emission (Fangmeier et al., 1994). In addition, ammonia reacts with sulfuric ($H_2SO_4$) and nitric acid ($HNO_3$) in the atmosphere, leading to the transformation from ammonia to fine ammonium salts (($NH_4)_2SO_4$, $NH_4HSO_4$, $NH_4NO_3$) (Schaap et al., 2004). The ammonium salts account for a large fraction of particulate

matter which has a longer lifetime in the atmosphere and is subject to long-range atmospheric transport (Fowler et al., 2009). Particulate matter has various negative societal impacts. It is a major contributor to smog and is associated with severe negative effects on human health (Brunekreef and Holgate, 2002; Pope et al., 2009). Furthermore, they influence the scattering of sunlight, alter the number, size and hygroscopic properties of cloud condensation nuclei, causing visibility impairment and disturbing the radiance balance of the Earth (Charlson et al., 1991; Erisman et al., 2007). Once deposited, the nitrogen

components can lead to acidification and eutrophication of ecosystems, which will result in loss of biodiversity (Bobbink et al., 2010; Krupa, 2003; Vitousek et al., 2008).

Even though ammonia emissions contribute to a range of threats to the environment and human health, there are large uncertainties in ammonia budget and its distribution at global and regional scale, illustrated by errors of more than 50% (Erisman et al., 2007; Sutton et al., 2014). Ammonia emissions from agricultural activities are prone to considerable spatial

and temporal variability (Battye et al., 2003; Sutton et al., 2003). Emissions from some activities are short term and highly variable, such as manure and fertilizer application, while some other activities contribute long term and less variable emissions, such as animal housing and manure storage. Many factors influence the variability of agricultural ammonia emissions (Battye et al., 2003; Dennis et al., 2010; Hutchings et al., 2012; Pinder et al., 2004, 2006), including:

- Local agricultural practices

30       o   Type and amount of manure and inorganic fertilizer applied to the land
        o   Method of manure and fertilizer application
        o   Animal type, housing type, manure storage type
- Meteorological conditions (air temperature, wind speed, humidity)



- Soil conditions (soil temperature, texture)
- Regulation of agricultural practice

Several emission inventories have been developed to improve the spatial details of ammonia emission in different countries. In MACC-III, emission factors and proxy maps are utilized to obtain the spatial distribution of annual emissions from emission totals officially reported by countries (Kuenen et al., 2014; Velthof et al., 2012). Hutchings et al. (2001) introduced a nitrogen flow approach to model annually averaged $NH_3$ emission for Denmark, taking into account animal types or different amount of fertilizers applied on various regions. In their study, ammonia emissions are calculated as a percentage of the total N in manure, which means that the model will be valid as long as the chemical and physical characteristics of the manure remain the same. It also indicates that the model can be easily adapted as long as the only parameters that change are the number of animals or their distribution between the manure handling systems. Similar methodology has been adopted by Gac (2007) in France, Webb and Misselbrook (2004) in the UK and Hyde (2003) in Ireland.

Subsequently, temporal distribution profiles are utilized to obtain temporally resolved emissions. Skjøth (2004) demonstrated an implementation of a simplified version of the dynamic parameterization for Denmark in the air pollution model ACDEP, by correlating temperature with emission functions or 15 agricultural subsectors, taking into account physical processes like volatilization as well as agricultural production methods including the timing of fertilization. Significant improvements have been witnessed compared to the results obtained by utilizing simplified time profile for agricultural emissions. Based on the work of Skjøth (2004), Gyldenkærne (2005) improved the parameterization by including the effect of ventilation rates inside buildings, ambient wind speeds and a more realistic description of temperatures inside animal houses.

Current emission inventories used in European chemistry transport models (CTMs) usually distinguish sectors defined by EMEP SNAP level 1 categorization, which has a single sector for agriculture. They do not indicate crop types and fertilizer types that are important for the interpretation of the results and future application of the model such as policymaking. Furthermore, in most European regional scale CTMs, such as LOTOS-EUROS (Hendriks et al., 2016; Schaap et al., 2008), the accompanying time profiles that allocate gridded emission in time are mostly generated by simplified and static seasonal functions, without taking into account local climatology and agricultural practices. However, it is a challenge to improve this situation for European scale applications as detailed ammonia emission modeling requires detailed information about agriculture activity data and the spatial distribution of farmhouses, storages, and number of different livestock as well as cropland types (Skjøth et al., 2004; Gyldenkærne et al., 2005).

In view of the above shortcomings, we developed a novel ammonia emission model that quantifies agricultural emissions with better spatial details and gives insight into the temporal dynamics. The improvement of the spatial emission allocation was realized by embedding the results of the INTEGRATOR model into MACC-III emission inventory. INTEGRATOR assesses greenhouse gases and nitrogen fluxes from agricultural sectors at high spatial resolution and accounts for differences in crop types, fertilizer types, animal housing and manure storage (De Vries et al., 2011; Kros et al., 2018). More detailed temporal distribution came from the emission functions from Gyldenkærne et al. (2005) and Skjøth et al. (2004) with the integration of the TIMELINES model which provides predictions of timelines of key agricultural operations across Europe (Hutchings et al.,



2012). These new emission data were then used in LOTOS-EUROS for verification by comparing modeled total columns and surface concentrations with measurements. In this work, the improvements of ammonia emissions were made for Germany and Benelux in the year of 2010 as a first test case.

In this paper, we first describe the methodology of 1) the new emission model which generates spatially and temporally
resolved emission products; 2) the chemistry transport model LOTOS-EUROS that translates emission into concentrations and total columns; 3) data processing of the available measurements for comparison and validation. Then we visualize the simulated results obtained from the original and updated ammonia emission model and evaluate the performance by comparing modeled total columns and surface concentrations with remote sensing and ground-based observations. Finally, we clarify in which way the model has been improved and point out the shortcomings of the updated model for future perspectives for this
work.

## 2 Methodology and Data

A schematic overview of the methodology and workflow is presented in Figure 1. The new emission model is composed of two parts, a spatial allocator which produces gridded maps of ammonia emissions from various categories and a temporal allocator that disaggregates the annual emission within a grid cell over the course of a year, creating emission distributions in
space and time. The spatial allocator integrates the detailed agricultural emission information from INTEGRATOR in MACC-III. The temporal allocator, with the help of the agricultural management model TIMELINES, characterizes the temporal variation according to land use, agricultural practice and climate, and translates the annual emission per grid cell and category into hourly time series. The emission estimates were then imported into the CTM LOTOS-EUROS to derive ammonia concentrations which were subsequently compared with IASI (Infrared Atmospheric Sounding Interferometer) observations
on ammonia total columns and in-situ measurements of surface concentrations for verification. Normalized root mean square error (NRMSE), normalized mean absolute error (NMAE), model efficiency (EF) and index of agreement between estimates and measurements were calculated to determine the performance of the models (Appendix A).

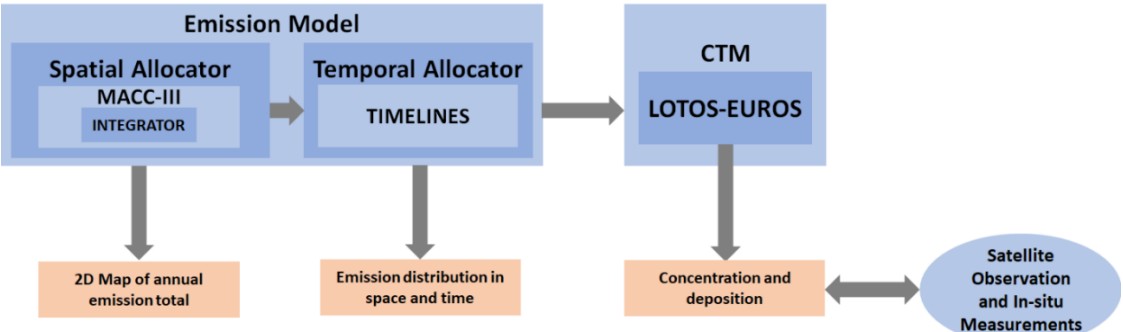

**Figure 1 A simplified scheme of the workflow in this project, involving the development of spatial and temporal allocators of the emission model as well as verification with measurement data.**





## 2.1 Model Parameters

In this study, we focused on ammonia emission estimates within the region containing Germany and Benelux for the year of 2010. Therefore, the spatial domain of the area of interest was set to be $2°E \sim 16°E$ in longitude with a step of 0.125 degrees, and $47°N \sim 55°N$ in latitude with a step of 0.0625 degrees, which corresponds to 112 pixels and 128 pixels in longitude and latitude direction, respectively. Two simulation runs were performed to identify the influence brought by the new method. In the first simulation, the original MACC-III annual emission distribution and LOTOS-EUROS time profiles were used. The second simulation run utilized the improved spatial distribution and the dynamic time profiles obtained with the updated model.

## 2.2 Spatial Allocator

### 2.2.1 The MACC-III inventory

MACC-III (Modeling Atmospheric Composition and Climate) is a spatially explicit emission inventory with a resolution of $0.125° \times 0.0625°$ longitude-latitude (approximately $7km \times 7km$), providing Europe-wide annual emission inputs for $NO_X$, $SO_2$, $NMVOC$, $CH_4$, $NH_3$, $CO$, $PM_{10}$ and $PM_{2.5}$ for air quality models (Kuenen et al., 2014). The inventory is based on national emission totals per sector officially reported by the countries themselves. In case emission data for a sector/country are unavailable for a certain year, estimates from GAINS are used to make sure that the emission inventory is complete and applicable for every country in Europe (Kuenen et al., 2011). Emission totals are spatially disaggregated across the countries in the form of point or area sources, using point source locations and proxy maps (e.g. population density, traffic intensity), respectively (Kuenen et al., 2014). Due to the top-down nature of the inventory, it does not distinguish agricultural ammonia emission distributions from types of animal housing and manure storage, application of various fertilizers on croplands and grassland. Instead, it differentiates emissions by animal types which include the application and storage of certain animal manure and housing of this animal, as well as the application of mineral fertilizer.

We aimed to improve the inventory towards a more detailed categorization so that it can provide more in-depth information on the impact of various agricultural activities on emission. In addition, the available information in the inventory does not fulfill the requirements of the TIMELINES model. The disadvantages mentioned above are the reason why we introduced the INTEGRATOR model in this study.

### 2.2.2 The INTEGRATOR Model

The INTEGRATOR model (Integrated Nitrogen Tool across Europe for Greenhouse gases and Ammonia Targeted to Operational Responses), is a static N cycling model that is used to calculate land system budgets at EU 27 level including N uptake, N emissions (in the forms of $NH_3$, $N_2O$, $NO_X$ and $N_2$) from housing and manure storage systems, N accumulation in or release from the soil (due to manure and mineral fertilizer application) and N losses by leaching and runoff (De Vries et al., 2011), based on the MITERRA model (Velthof et al., 2007, 2009). INTEGRATOR is an adapted, more detailed version of the former MITERRA-Europe model. The emissions of ammonia and other gases ($N_2O$, $NO_X$ and $N_2$) to the atmosphere are





estimated by multiplying N inputs with emission factors (De Vries et al., 2011). In this study, we focus on the modules of the model that estimate ammonia emissions from animal housing and manure storage systems as well as the application of manure and mineral fertilizer to arable land and grassland.

Unlike the MACC-III inventory which provides emission distributions on longitude-latitude grids in World Geodetic System (WGS84), INTEGRATOR estimates emissions in NitroEurope Classification Units (NCUs). These NCUs are multi-part polygons composed of a number of 1 km × 1 km grid cells in ETRS89/LAEA Europe coordinate system in the domain of EU. The polygons sharing one NCU number have the same administrative unit (NUTS2 or NUTS3), soil type (SGDB classification), similar slopes (CCM DEM 250 in five classes) and altitude (with differences less than 200m) (De Vries et al., 2011). Therefore, the area of one NCU varies from several square kilometers (mostly in Western and Southern Europe) to hundreds of square kilometers (in Northern Europe).

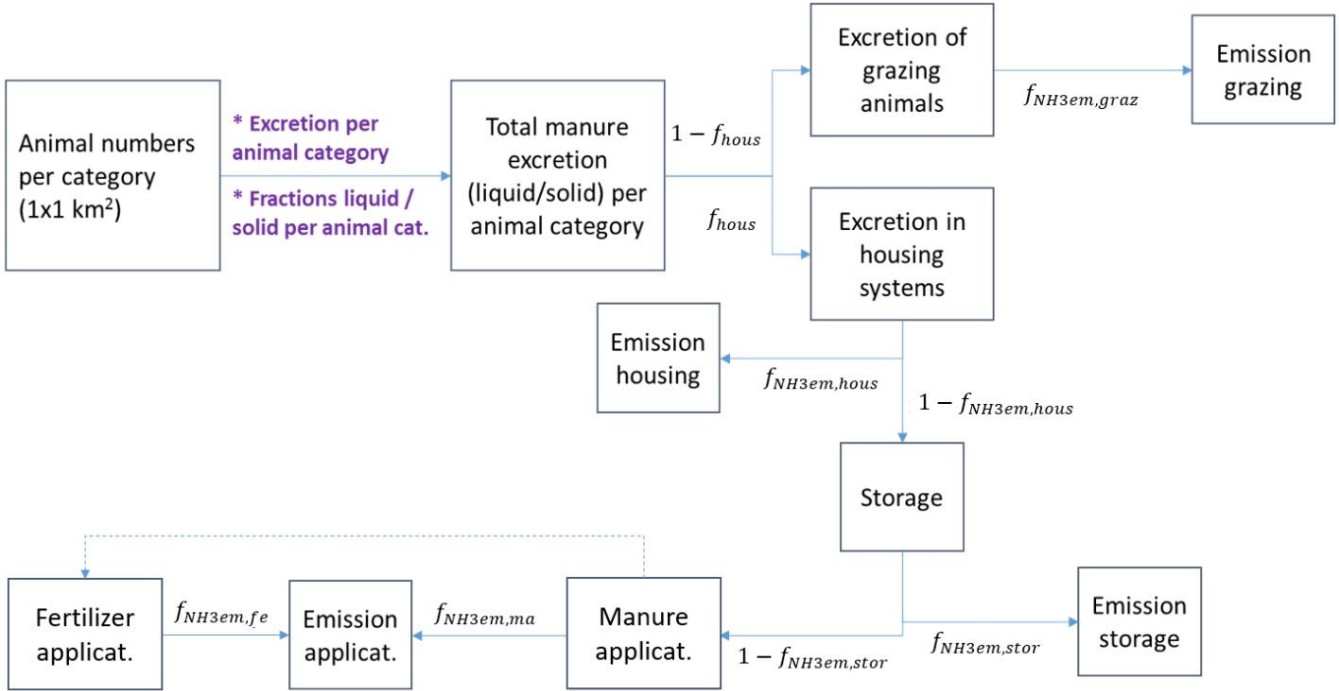

**Figure 2 A schematic workflow of the ammonia emission module in INTEGRATOR. $f_{hous}$ is the fraction of total manure excretion going to housing systems. $f_{NH3em,graz}$, $f_{NH3em,hous}$, $f_{NH3em,stor}$, $f_{NH3em,ma}$ and $f_{NH3em,fe}$ represent emission fractions of grazing, animal housing, manure storage, manure application, and fertilizer application, respectively.**

A schematic overview of the ammonia emission module of the INTEGRATOR model is presented in Figure 2. The emission model starts with the calculation of N excretion by multiplying the number of animals at NCU level with N excretion rates per animal per country for eight animal categories (dairy cows, other cows, pigs, laying hens, other poultry, sheep and goats, horses and fur animals). The livestock data are obtained from FAO database at country level, using CAPRI data for distribution at



NUTS 2 level. The data on livestock numbers of various animal categories at NUTS2 level have been downscaled to a 1 km$^2$ resolution using expert-based judgment with spatial data sources on land use, slope, altitude and soil characteristics influencing the livestock carrying. A major distinction was made between grazing animals and other animals. Dairy cows, beef cattle, sheep and goats were assumed to be highly dependent on local land resources for grazing or feed production. Pigs and poultry

were assumed to be held in more land independent systems. For more detailed information on the downscaling of livestock, we refer to Neumann et al. (2009). The N excreted in housing systems is the multiplication of N manure excretion and the housing fraction ($f_{hous}$ in Figure 2), while the N excreted from grazing on land is obtained by subtracting N excreted in housing systems from total N manure excretion. The total manure production is derived by subtracting gaseous emissions and leaching in housing and manure storage systems from the N excretion, while the gaseous emissions are calculated by multiplying N

excretion with the emission fraction per housing system ($f_{NH3em,hous}$). Ammonia emission fractions for housing and manure storage are distinguished per animal type (6 categories) and manure type (liquid vs. solid for 3 animal categories). For some countries, the basic emission fractions are modified based on the degree of implementation of emission-reducing technologies, and the reduction efficiency of the technology (De Vries et al., 2020). The emissions of ammonia from agricultural land are calculated by multiplying the N input by grazing, manure application and fertilizer application with ammonia emission

fractions for grazing $f_{NH3em,graz}$), manure application ($f_{NH3em,ma}$) and fertilizer application ($f_{NH3em,fe}$), respectively. Emission fractions for manure application are distinguished for three animal types, i.e., cattle (dairy cows, other cows, sheep and goats, horses and fur animals), pigs and poultry (laying hens, other poultry) and manure type (liquid vs. solid) whereas emission fractions for fertilizer application are differentiated between urea-based fertilizers and nitrate-based fertilizers. Details on the various fractions are given in De Vries et al. (2020). As the last step, mineral fertilizer and manure are distributed over

crops on country level using a balanced N fertilization approach:

1.  The total N demand in a NUTS 2 region is calculated by multiplying the N uptake of each crop by the total area of the crops in each NUTS 2 region. The N removal in each crop is calculated as the product of the crop yield (in terms of harvest) and the N content in harvested crops.

2.  The available manure is evenly divided over the crop types according to their N demand. For example, when the

available manure in a region satisfies 80% of the total N demand, each crop gets its 80% share. Also, crop residues were accounted for to calculate the amount of effective non-fertilizer N input.

3.  The fertilizer N demand of each crop was calculated by subtracting the non-fertilizer N input from the total N demand and then divided by the N use efficiency (NUE).

4.  The N fertilizer estimates for each NUTS 2 region were aggregated at country level and compared with reported

country-level N fertilizer consumption. Scaling factors (the ratio of the known and calculated country-level N fertilizer consumption) were then applied to ensure consistency.

As a result, ammonia emissions in each NCU are available for 43 croplands (42 CAPRI crop types plus grassland) differentiating among 5 manure types (poultry, cattle liquid/solid, pig liquid/solid) and mineral fertilizer application, as well as from grazing, housing of three animal types and manure storage of 5 manure types, in total 267 categories.





### 2.2.3 The MACC-INTEGRATOR Combined Inventory

MACC-III provides the spatial distribution of annual ammonia emissions from agriculture and non-agricultural sectors including traffic and industry. We replaced the agricultural emissions in the original MACC-III inventory with the INTEGRATOR emissions, which significantly increases the level of details. As is demonstrated in Fig. B1 in Appendix B, for

simplification, the CAPRI crop types in INTEGRATOR were aggregated using the Indicative Crop Classification (ICC), into cereals, root crops, industrial crops, vegetables, grass and fodder. Consequently, there were 36 categories regarding emissions from fertilization on croplands. Grazing (1), animal housing (3), and manure storage (5) were kept as they were, resulting in 45 categories in total in the combined emission inventory.

Since MACC-III and INTEGRATOR estimate emissions at different coordinate systems, coordinate transformation was

performed to resample INTEGRATOR emissions onto the grid (WGS84) utilized in MACC-III. The resampling was conducted by 1) averaging the emission in one NCU evenly over the whole polygon; 2) dividing each square kilometer grid cell into 25 subpixels and calculating the coordinate of the center of each subpixel in latitude/longitude; 3) locating the calculated coordinate of each subpixel of NCU in MACC-III grid and assigning emission to the corresponding MACC-III grid.

It has to be noted that the ammonia emission estimates from INTEGRATOR differ from the officially reported national

emission totals which are used in the MACC-III inventory. Because each country utilizes its own estimation algorithms that deviate from the INTIEGRTOR methodology which starts with animal number, excretion rate and emission fraction. To assess the impact of the different spatial (and temporal) allocation and be in line with officially reported emissions, we scaled the ammonia emissions of INTEGRATOR to the country totals of 2010 officially reported in 2018. The scalar is computed per country and animal type, namely the division of INTEGRATOR emission and officially reported emission to EMEP.

## 2.3 Temporal Allocator

The usual approach to characterizing the temporal variability in ammonia emissions is to use time profiles that distribute annual emission total in a grid cell over the course of a year. Fixed and oversimplified temporal profiles (monthly, daily, or hourly resolved) are often used(Van Pul et al., 2009). In this section, we outline how we explicitly described the temporal allocation of ammonia emissions from manure and fertilizer application, grazing, animal housing and manure storage based on the

concepts of Skjøth et al. (2004), Gyldenkærne et al. (2005) and Hutchings et al. (2012).

### 2.3.1 Manure and fertilizer application on arable lands

The timing of manure and fertilizer application on arable lands and subsequent ammonia emission is dependent on the timing of manure and fertilizer application on croplands, weather conditions, as well as legislative constraints. We first followed the methodology as outlined by Gyldenkærne et al. (2005) to characterize the temporal variation of the emission strength as a

function of time, temperature, and wind speed. The emission function used may be described as Eq. (1):

$$E_{i,j,k}(t,T,W) = \epsilon_{i,j,k} e^{0.0223T(t)} e^{0.0419W(t)} \frac{1}{\sigma\sqrt{2\pi}} e^{\left(\frac{(t-\mu)^2}{-2\sigma^2}\right)} \qquad (1)$$





where $E_{i,j,k}$ is the emission strength after application of fertilizer $k$ on crop $j$ in NCU $i$, $\epsilon_{i,j,k}$ is the annual total emission (kg/ha) of fertilizer $k$ on crop $j$ in NCU $i$, T(t) and W(t) are the air temperature (Celsius) and wind speed (m/s) for the applied time step (t), $\mu$ is Julian day with the peak emissions, and $\sigma$ is the standard deviation to represent spread and uncertainty in the application activities and emission timing.

*The improvement of emission function*

The first challenge was to update the estimated central days (the time with peak emissions) for manure and fertilizer applications. The timing of these field operations was calculated by using the methodology of the TIMELINES model (Hutchings et al., 2012) that was developed to assess the timing of field operations in Europe at the 50 km × 50 km MARS meteorological grid level (Goot, 1997), including the Julian day of fertilization, for a wide range of crops. Hutchings et al.

(2012) took the weather conditions over a year into account when simulating crop calendars by introducing a thermal time approach. Thermal time is the sum of the positive differences between daily mean air temperature and a base temperature and is therefore written as Eq. (2):

$$\tau_t = \sum_{k=t_0}^{t} \max\left((\theta_k - \theta_b), 0\right) \qquad (2)$$

where $\tau_t$ is the thermal sum (in Celsius) over time t (day), $\theta_k$ is the daily mean air temperature at 2 meters, $\theta_b$ is the base

temperature (0 degree Celsius), $t_0$ is the starting time of calculation 1 January. As soon as thermal time on Julian day t reaches the reference thermal time for sowing (or harvesting), it is considered that sowing (or harvesting) most probably occurs. This approach is based on the assumption and gross simplification that the sowing and harvesting dates of crops can be related to accumulated air temperature, and that these two events can be used to frame all other field operations including plowing, N fertilization, and manure operations. More specifically, this logic assumes that farmers time fertilizer and manure application

to maximize nitrogen use efficiency for crop production. In general, the timing of fertilizer and manure application depends on the sowing date. Except for applications of mineral fertilizer and animal slurry to winter crops, the timing is related to the start of the growing season (Hutchings et al., 2012).

We back-calculated the reference thermal times based on the sowing and harvesting dates provided by Hutchings et al. (2012) and ECMWF meteorological data for the years between 1985 and 1995 by inserting the respective days $t_{sow(harv)}$ into Eq.

25    (3):

$$\tau_{ref,sow(harv)} = \sum_{k=t_0}^{t_{sow(harv)}} \max\left((\theta_k - \theta_b), 0\right) \qquad (3)$$

The period between 1985 and 1995 was selected as Hutchings et al. (2012) followed a similar proceeding based on the CGMS dataset and used obtained reference thermal times to calculate sowing and harvesting days for 1995 onwards. The sowing and harvesting dates derived in this paper are in good alignment with the work of Hutchings et al. (2012). An example is given in

Fig. C1.

Two examples of TIMELINES output are shown in Fig. D1 which demonstrates the Julian days of sowing time of winter wheat and spring wheat in 2010. In general, the sowing days of these two crops have the opposite trends. For winter wheat,





sowing occurs in Southern Europe later than it does in the north. Even though the difference between daily mean temperature and the base temperature is larger in the south, the greater reference thermal sum makes it longer to reach. Whereas for spring wheat, the reference thermal sum in the south is less significantly bigger than that in the north, resulting in earlier sowing day in the south.

The timing of manure applications is based on sowing dates and varies from one manure type to another. Hutchings et al. (2012) assumed that applications of solid manure to both spring and winter crops are placed five days prior to the sowing date. Applications of animal slurry for spring crops coincided with the application of solid manure, while for winter crops the applications are put at the start of the growing season. The start of the growing season for the winter crops at a given location is equated to the sowing date for spring barley at the same location, and the end is the sowing date for winter wheat.

The timing of fertilizer applications is designed to promote efficient use of the fertilizer N; the annual amount is applied in two applications. The first application consisting of 20% of the annual amount is conducted five days prior to sowing for spring crops and at the start of the growing season for winter crops. The second application, which is composed of the remaining 80% of the annual amount, is made after 20% of the growing season has elapsed. The growing season for spring crops is from the sowing date to the harvesting date. Then the timing was subsequently modified to ensure that the second fertilizer application

did not take place within 21 days of harvesting (Hutchings et al., 2012).

Even though the TIMELINES model indicates a single day of fertilization in an NCU, in practice, farmers certainly would not operate precisely at the same time. The central estimate of fertilization day is uncertain due to other influencing parameters such as soil conditions as well as the availability of machinery and labor. Also, Gyldenkærne et al. (2005) argued that there still expect to be variation in the timing of fertilizer and manure application because farmers often spend several days applying

fertilizer and manure to the field. This means that a normal distribution around the central estimate and Gaussian functions are used to characterize it.

The standard deviation around the central value is given in a fixed number of days since it is determined by the agricultural practice of farmers (independent of the thermal sum approach) and includes a random uncertainty. Gyldenkærne et al. (2005) assumed there are four times of manure application in a year: early spring, late spring, spring-summer, summer-autumn. Except

that spring-summer application's deviation is 16 days, the other applications are given a deviation of 9 days. Besides, mineral fertilization in early spring and summer have a deviation of 9 and 16 days, respectively. Therefore, in this paper, we followed the systematic: for fertilizations that lie between mid-May and mid-August the deviation of corresponding emission function is 16 days, while for the others the standard deviation is considered to be nine days.

The relationship between the timing of peak emission after manure and fertilizer application has been studied. Field

experiments show that the emission from mineral fertilizers has its maximum in the first days after application (Loubet et al., 2009; Schjoerring and Mattsson, 2001; Whitehead and Raistrick, 1993). Søgaard et al. (2002) observed that half of the $NH_3$ emission takes place within the first 30 hours. Plöchl (2001) looked into 227 experimental trials and found that 80% of the emission was reached within 2 days. However, in some cases (e.g., urea applied in dry conditions resulting in slow hydrolysis), fertilizer emission may proceed for over a month after application (Sutton et al., 1995). As these dry conditions are unlikely in





our study area, there no reason to assume a large delay between application and emission. We assumed that the peak of emission after application occurs at noon on the second day after the estimated central fertilization day. This factor also accounts for the fact that farmers are more likely to delay than advance activities. Hence, the width around the mean discussed above accounts for both uncertainties in fertilization day estimates ($T_{sum}$) and time scale of emission. Because the timing of fertilizer

application and that of manure have more or less the same limitations, it could be assumed that they have the same pattern in this case for simplification.

*The inclusion of legislative conditions*

The next step is to implement legislative constraints on the variability of manure and fertilizer application. In Germany, there should not be manure spreading from 1 November to 31 January on arable land, and from 15 November to 31 January on

grassland (Kuhn, 2017). In Flanders, Belgium, manure spreading is not allowed in the winter period from 15 October till 15 February (Vlaamse Land Maatschapij, 2016b), which is expanded to the whole of Belgium and Luxemburg in this study due to lack of knowledge in these regions. As for the Netherlands, solid manure is prohibited from 1 September to 31 January, while other manures are banned between 16 September and 15 February on arable land and between 1 September and 15 February on grassland. Mineral fertilizer is prohibited from to 16 September to 31 January on both grassland and arable land

(RVO, 2019). Furthermore, (Vlaamse Land Maatschapij, 2016a) pointed out that in Flanders, it is not allowed to fertilize where the soil is frozen or covered by snow, during the period with vegetation calm, and on Sundays. Frozen soil, snow cover, and vegetation calm commonly occur before the start of the first spring application distribution. Also, they tend to disappear together before spring application because their disappearances all need a few days with above normal temperature conditions, mostly associated with precipitation. As a result, the introduction of the ban on fertilization outside permitted dates and on

Sundays is the most influential constraint and applied to all regions in the area of interest by setting the emission strength to zero.

*The impact of excessive precipitation*

Another factor that needs to be taken into account is precipitation. When there is excessive precipitation, the soil becomes water-saturated, which will have a negative impact on the intrusion effectiveness of manure and lead to an enormous amount

of nutrient losses. Besides, after excessive precipitation, it is much more difficult for farmers to perform fertilization practices. Therefore, manure application is not effective during these conditions. We introduced De Martonne-Index to capture the characteristics related to precipitation or soil water content. The index describes the ratio between precipitation sums and average 2-meter temperature and may be calculated both on annual and on a shorter period basis (Croitoru et al., 2012). For annual values, it is written as Eq. (4):

$$I = \frac{P}{T+C} \qquad (4)$$

where P is annual total precipitation in millimeter, T is annual mean temperature in Celsius, C is a constant that assures that negative mean temperatures do not result in negative indices and is equated to 10. Instead of only utilizing precipitation information, the introduction of temperature parameterizes the impact that higher temperature will lead to faster evaporation





and more effective infiltration. Here, the index is computed on weekly basis to represent more real-time humidity. Similar to the monthly and seasonal De Martonne-Index, weekly De Martonne-Index can be written as Eq. (5):

$$I_w = \frac{52.143 P_w}{T_w + C} \qquad (5)$$

where $P_w$ is weekly total precipitation in millimeter, $T_w$ is weekly mean temperature in Celsius. Here the constant 52.143 is
neglected for simplification. Baltas (2007) defined that when annual De Martonne-Index exceeds 55, in our case weekly index exceeds 1.055, the air is considered extremely humid. One example is given in Fig. E1. Together with visual inspection, a threshold of 1.7 is set, above which precipitation and soil water content are not suitable for fertilization, and farmers will have to postpone application. Therefore, for each day on which the threshold is violated, manure application is set to zero, and the remaining part of the normal distributions is moved forward by a day.

*Finalization of emission time profile*

From above, the temporal variation of emission strength for application of fertilizer $j$ on crop $i$ in NCU $n$ is derived. The ammonia emission time profile needed by the LOTOS-EUROS model has an hourly temporal resolution and a mean of 1. However, before normalizing the emission strength, a baseline in the time profile is introduced. Due to various fertilizer application techniques, especially in the case of injection, manure and fertilizer stay underneath the soil for a much more
extended period before ventilation. Thus, 5% of the profile sum is allocated throughout the year as a baseline to represent background emission. Afterwards, emission strength variation is normalized by the remaining 95% of the profile sum and added to the baseline.

Examples of ammonia emission time profiles during construction at location $(47.41°N, 10.98°E)$ in latitude/longitude in 2010 are presented in Figure 3. The left panel represents time profiles of the application of cattle liquid manure on cereals, while the
right panel demonstrates that of pig liquid manure application on grass and fodder. Four rows indicate the four phases during the development of the time profiles. In both panels, first and foremost, initial emission time profile (first row) is obtained using fertilization day estimation from TIMELINES and emission function Eq. (1) from Gyldenkærne et al. (2005), taking into account local climatology including temperature and wind speed. Subsequently, the emission factors of Sundays are set to baseline since manure and fertilizer application are assumed to be prohibited, as is shown in the second row. Moreover, in the
third row, prohibition on fertilization after late fall and before early spring (exact dates vary from country to country) does not affect the time profile on the left panel since the emission function lies within the period where fertilization is allowed. However, for the right panel, part of the third peak extends further than the last allowed date for application. Thus, the part outside the boundary is cut out from the curve, and the rest of the peak is scaled to ensure that its sum is kept unchanged. Finally, the impact of excessive rain on emission is accounted for in the last row. On each day where the De Martonne-Index
exceeds threshold 1.7, the part of the emission curve before this day remains as it is, while the rest is shifted to the next possible day. It is possible that emission lies outside the permitted period for fertilization, but we assume the government allows the condition where manure and fertilizer application have to be delayed due to weather.



Finally, the time profile for application of fertilizer $j$ on crop $i$ in a standard grid cell is derived by resampling using area averaging. In comparison with the original time profile used in LOTOS-EUROS, the newly developed time profile is spatially and dynamically explicit based on land type, amounts of emission and local climatology, unlike the original profile which is only country dependent, and indistinguishable among different agricultural sectors.

**Figure 3 Two examples of ammonia emission time profile during the four phases of development at location (47.41, 10.98) in latitude/longitude. Left and right panels represent two sectors at the same location, with left being cattle slurry application on cereals, and right being pig liquid manure application on grass and fodder**





### 2.3.2 Grazing and fertilization on grassland

For the temporal variation of ammonia emission from fertilization on grassland, we used the parameterizations of Skjøth et al. (2004) for Danish conditions using a gauss-function as given below:

$$
\begin{cases}
F_{grass} = E(x,y) \times e^{0.0223T(t)} e^{0.0419W(t)} \times \dfrac{e^{\left(\frac{(t-\mu)^2}{-2\sigma^2}\right)}}{\sigma\sqrt{2\pi}} \\
\mu = T_{sum1400}(x,y) + 4
\end{cases}
\tag{6}
$$

where $t$ is the actual time of the year, $E(x,y)$ is the total emission from fertilization on grassland within a grid cell, $\mu$ is the mean value for the Gaussian distribution, $\mu$ depends on local climatology thus it differs from grid cell to grid cell, T(t) is the air temperature in Celsius, W(t) is the wind speed (m/s) for the applied time step (t). $\mu$ is the Julian day on which thermal sum reaches 1400, except that the starting day of thermal time calculation is 1$^{st}$ March, instead of 1$^{st}$ January. $\sigma$ is the spread of the gauss function and is equated to 60 days, which means that grazing occurs in a relatively long period of time.

Regarding emissions from grazing on grassland, generally it is dependent on the release time of the cattle, the availability of grass, and the length of the growing season (Gyldenkærne et al., 2005). The availability of grass is then primarily a function of precipitation, soil humidity, soil fertility, and fertilization. For a region that has a relatively even distribution of the precipitation during summer, such as the study area in this paper, Gyldenkærne et al. (2005) suggested that a model following grass growth could be used to represent the characteristics of grazing emissions. Therefore, as the work of Skjøth et al. (2004),

here emission from grazing is assumed to follow the same pattern as grown grass in Eq. (6).

### 2.3.3 Animal housing and manure storage

Emission patterns from animal housing and manure storage are based on Skjøth et al. (2011) and Gyldenkærne et al. (2005) as given below:

$$
\begin{cases}
Fkt_i = \dfrac{E_i(x,y)}{Epot_i(x,y)} \times (T_i(x,y))^{0.89}, & T_i(x,y) \geq T_{boundary} \\
T_i(x,y) = \begin{cases} 18 + 0.77 \times (T(x,y) - 12.5), & \textit{Insulated houses} \\ T(x,y) + 3, & \textit{Open houses} \\ T(x,y), & \textit{Manure storage} \end{cases}
\end{cases}
\tag{7}
$$

where $i$ refers to the index (1-3) of insulated housing, open housing and manure storage, respectively. $x, y$ are the coordinates of the emission grid. $E_i(x,y)$ represents the emission for the corresponding agricultural sector within the grid cell. $Epot_i(x,y)$ is a constant emission potential scaling factor for a given grid cell and can be neglected for simplicity (Elzing and Monteny, 1997). $T_i(x,y)$ is temperature function which is different for housing, open housing and manure storage. $T(x,y)$ is the 2-meter temperature at the given location and is obtained from the ECMWF data portal. It can be seen from Eq. (7) that open houses

and manure storage have almost the same emission pattern except that the indoor temperature in open houses is 3 degrees higher than the outside temperature used for manure storage (Gyldenkærne et al., 2005). $T_{boundary}$ represents lower boundary condition for temperature in animal housing and manure storage, below which emission is set to a constant level, and they are 18, 4, and 1 degree, respectively.





Pigs and poultry have a high lower critical temperature (LCT) between 6 to 20 degrees, so in colder climates, they are usually kept in insulated buildings with forced ventilation (Seedorf et al., 1998b) to maintain a fixed temperature throughout the year. On the contrary, cattle have a very low LCT and thus often kept in open barns (Seedorf et al., 1998a). However, there still might be some insulated cattle barns with forced ventilation in colder climates (Gyldenkærne et al., 2005). Consequently, the

function for forced ventilation is used to represent the temporal variation of pig and poultry housing emission, while the mean of functions of insulated houses with forced ventilation and open houses is calculated to characterize cattle housing emission. In terms of manure storage, it is assumed that the emissions from manure storage of all animal types have the same pattern.

## 2.4 The LOTOS-EUROS Model

After spatial and temporal allocation of ammonia emission, the annual emission distribution and gridded hourly time profile

are imported into the chemistry transport model LOTOS-EUROS to obtain modeled surface concentrations and total columns for model evaluation by comparing simulated results with satellite observations and in-situ measurements. LOTOS-EUROS is a 3-dimensional regional CTM that uses a description of the bidirectional surface-atmosphere exchange of ammonia (Kruit et al., 2010; Manders et al., 2017). In the previous studies, the model showed a good correspondence with yearly averaged ammonia measured concentrations, e.g., slightly underestimating concentrations in agricultural source areas and slightly

overestimating concentrations in nature areas (Wichink Kruit et al., 2012). In the older version of LOTOS-EUROS, the temporal variation of ammonia emissions is represented by simplified monthly, day-of-the-week and hourly time factors, which allows distinguishing different countries, but does not account for different agricultural categories. The version of LOTOS-EUROS in this study included the labeling module by Kranenburg et al. (2013), which tracks the contribution of emission sources from specific categories to the final simulated products, such as surface concentration and 3-dimensional

concentration. The categories that we want to label, namely all agricultural sectors, were defined accordingly before the model runs. As a result, besides the regular outputs, the fractional contribution of each labeled category was also calculated.

## 2.5 Available Measurements

Among the outputs of LOTOS-EUROS, surface concentration and 3-d concentration could be compared with in-situ measurement and satellite observations for verification. Both in-situ and satellite observations have their advantages and

disadvantages. The transport of ammonia in the atmosphere and the reaction with other atmospheric components are rapid, leading to the fact that its emission and deposition dynamics affect concentrations on the scale of hours to days.

Ground-based stations measure ammonia surface concentration level consistently at fixed locations and some of them have relatively high temporal resolutions (hourly or daily), offering the possibility to study the behavior of ammonia emission. However, the measurements lack vertical information and horizontal representation and most instruments only measure surface

concentrations (Van Damme et al., 2015; Erisman et al., 2007). Airborne measurements have been carried out, but only occasionally with limited spatial coverage during campaigns (Dammers et al., 2016; Leen et al., 2013; Nowak et al., 2010). Horizontally, the setup of station networks is rather coarse, and representativeness of the observations is an issue since





measurements of all monitoring sites will be influenced by local and regional agricultural activities and other local sources. Consequently, we need to carefully take into account the locations of the stations when comparing in-situ measurements with simulated results.

Satellite observations have the advantage of global coverage and the possibility to calculate area-averaged observations which
are in much better correspondence with the size of the grid cells in regional/global models (Flechard et al., 2013). Remote sensing products with a higher spatial and temporal resolution have become available for better $NH_3$ concentration monitoring in the lower troposphere (Clarisse et al., 2009; Van Damme et al., 2015).With the launch of the latest instrument IASI, it is possible to achieve daily temporal resolution for ammonia monitoring, but it is still not sufficient for real-time monitoring. Also, measuring total column using satellite requires clear-sky conditions. Furthermore, the retrieval algorithm if IASI needs
an accurate temperature profile, without which larger measurement errors will occur.

### 2.5.1 In-situ measurements

The Umweltbundesamt (UBA) research foundation sets up monitoring stations, providing governments and the public with information on the concentration of air pollutants. It measures species, including ammonia, that are essential for the improvement of knowledge about air quality and climate change. The UBA also collects the data from the network of the
German federal states. In addition to the German networks, the Measuring Ammonia in Nature (MAN) network monitors monthly mean values of ammonia concentrations in Natura2000 areas in the Netherlands to detect the spatial pattern in concentration or to assess the influence of local sources (agriculture activities but also traffic) (https://man.rivm.nl/) (Lolkema et al., 2015).The network aims to be representative of different habitat types, ammonia concentration levels, area size and shape, as well as the geographical distribution (Lolkema et al., 2015). Even though for the comparison with modeled
concentrations, the used measurement data should be representative of a wider region, it is impossible to get rid of local influences completely. In this study, all measurements were also looked to determine the overall performance of the original and updated model. To illustrate the comparison for individual time series, we selected several stations (as shown in Table F1) so that the local influences on measurements could be minimized.

### 2.5.2 Satellite Observations

Infrared Atmospheric Sounding Interferometer (IASI) is a Fourier transform infrared (FTIR) spectrometer that measures the thermal infrared (TIR) radiation emitted by the Earth's surface and the atmosphere. It circles in a polar Sun-synchronous orbit and operates in nadir mode. It has a wide swath width of 2 x 1100 km, which corresponds to 2x15 mirror positions, while the spatial resolution is 50 km x 50 km, composed of 2 x 2 circular pixels. Each circular pixel is a 12 km diameter footprint on the ground at nadir (Clerbaux et al., 2009).
An improved retrieval scheme for IASI spectra was presented by Van Damme et al. (2014), which relies on the calculation of a dimensionless Hyperspectral Range Index (HRI). Whitburn et al. (2016) continued with HRI and introduced a neural-network-based algorithm to obtain ammonia total columns. Van Damme et al. (2017) made some improvements by training





separate neural networks for land and sea observations, reducing and transforming the input parameter space for enhanced thermal contrast, and introducing a bias correction over land and sea and the treatment of the satellite zenith angle, which resulted in the latest product Artificial Neural Network for IASI ANNI-NH3-v2.1. In this paper, we used a similar version of the IASI dataset ANNI-NH3-v2.2R-I which was obtained with meteorological data from ECMWF ERA-Interim and surface

temperature data retrieved from a dedicated network, instead of the operationally provided Eumetsat IASI Level 2 (L2) dataset used in the standard baseline version.

Regardless of the improvement of ammonia column retrieval from satellite observations, there is still a very large variability in measurement uncertainty, varying from 5% to over 1000 %, with summer daytime being the best time to measure ammonia while winter nighttime being the worst (Van Damme et al., 2017). Because thermal contrast, whose accuracy needs to be

improved, leads to the variable sensitivity of the outgoing infrared radiation to the lower troposphere (Clarisse et al., 2010; Van Damme et al., 2017). As a result, we only use satellite observations with daytime overpass in 2010.

Remote sensing data is sufficient if used to calculate monthly or yearly average distributions (Van Damme et al., 2017). In this study, the annual average was obtained to be compared with LOTOS-EUROS output for verification of spatial distribution, and the monthly mean was also calculated to investigate the feasibility of being used for temporal distribution validation.

LOTOS-EUROS simulations are required to be sampled and averaged in order to be harmonized for comparison with ammonia total column measurements. For each IASI observations, the modeled results that are closest in space and time are selected. Van Damme et al. (2014) considered weighted averaging, where the weight is inversely proportional to the square of the relative or absolute error for both model simulations and satellite measurements. Weighted averaging with relative error leads to a biased result towards overestimation while weighting with the absolute error favors the lowest columns. As is pointed out

by Van Damme et al. (2017), weighted averaging is no longer recommended with the extended post-filtering introduced in ANNI-NH3-v2.1. If possible, it is better to use individual measurements and avoid averaging. Arithmetic mean or median is suggested if averaging has to be performed.

After we obtained the dataset from the AERIS portal (https://iasi.aeris-data.fr/NH3R-I_IASI_A_data/), filtering was conducted to select valid measurements, which was based on cloud coverage (< 10%), column values (positive) and relative absolute

errors (<75%). Subsequently, area-weighted annual mean was obtained by re-gridding the footprints in the form of circular pixels onto the grid used in LOTOS-EUROS. Area averaging also applied to the calculation of averaged relative error of each grid cell. Finally, post-filtering was carried out to obtain more reliable distributions: all grid cells with less than ten measurements and a mean error larger than 75% for the morning orbit above land were rejected.

## 3 Results

### 3.1 Comparison between original MACC-III and updated MACC-INTEGRATOR annual emission

Because of the less detailed emission classification in the MACC-III inventory, comparisons of cattle, pig, poultry related annual emissions (the sum of housing, manure storage and application), as well as mineral fertilizer emissions, were made at



country level. Table 1 shows that country emission totals from the new inventory are all larger than those from the original MACC-III inventory due to different reported country totals used. Germany witnesses the largest positive difference in absolute value, while Luxemburg shows the biggest relative change. Compared to MACC-III, MACC-INTEGRATOR estimates more emission from cattle and fertilizer application for all countries except for the Netherlands. Pig emission in

Germany and the Netherlands from the updated inventory rises by 24.7% and 36.4%, respectively, while that in Belgium remains almost the same. Updated poultry emission is more than 20% lower in Germany, whereas the amount ascends in other countries. It implies that the scaling we utilized per country based on animal types and mineral fertilizer plays an important role in terms of emission country totals. For example, INTEGRATOR estimates less emission in Germany (as is mentioned in Sect. 2.2.3 The MACC-INTEGRATOR Combined Inventory), however after scaling with the 2018 EMEP reported national

emissions for 2010, the combined inventory reveals the opposite result which indicates 14% more emission in Germany than MACC-III.

**Table 1 Ammonia emission country totals (Gg/yr) for all agricultural categories and for cattle, pig, poultry and mineral fertilizer in the year 2010.**

|  | Germany | | Netherlands | | Belgium | | Luxemburg | |
|---|---|---|---|---|---|---|---|---|
|  | Original | Updated | Original | Updated | Original | Updated | Original | Updated |
| **Cattle** | 290.02 | 333.22 | 59.36 | 53.60 | 29.48 | 30.02 | 3.30 | 4.49 |
| **Pig** | 105.86 | 131.96 | 23.57 | 32.15 | 22.34 | 21.73 | 0.52 | 0.41 |
| **Poultry** | 46.62 | 37.11 | 14.11 | 19.58 | 4.41 | 5.30 | 0.04 | 0.08 |
| **Fertilizer** | 69.48 | 82.60 | 9.62 | 9.69 | 7.25 | 8.70 | 0.39 | 0.77 |
| **Total** | 513.05 | 584.89 | 106.70 | 115.03 | 63.97 | 65.76 | 4.26 | 5.72 |
|  |  | (+14%) |  | (+7.8%) |  | (+2.8%) |  | (34.3%) |

The spatial distributions of ammonia emissions from the two inventories are presented in Figure 4. Figure 4(a) and Figure 4(b) are the maps of annual total emissions. In Germany, the new spatial allocator assigns more emission in the southeast near the border with Austria. The two hot spots in Bremen and Ruhr in the original inventory merge into one, located in the Ostwestfalen-Lippe region. To the very north of Germany in Schleswig-Holstein, the original MACC-III indicates that most

of the emissions are concentrated in the middle of the state, while the updated one tells they are mostly situated along the coastline to the east. In the southeastern part of the Netherlands, the updated inventory allocates more emissions than the original one and smoothens the distribution details into larger blocks, which decreases spatial details. After looking into the shapefile of NCU, we found out the sizes of polygons at this location are quite large. MACC-INTEGRATOR calculates emission within an NCU polygon and distributes it evenly over the area, leading to the possibility of losing spatial details,

especially for a polygon with a larger size. Row 2-5 demonstrate the distribution of subsectors. Figure 4(c) and Figure 4(d) show that cattle emission estimate remains a similar pattern in the updated inventory, except that it is generally lower in





northwestern Germany and the Netherlands, leading to the disappearance of hot spots in the provinces of Overijssel and Gelderland in the east of the Netherlands. On the contrary, southern Germany, bordering Switzerland and Austria, witnesses more estimates in cattle emission. Figure 4(e) and Figure 4(f) illustrate that from the updated inventory, there appears to be more pig emission in the southeast of the Netherlands while it is more spread out in Nordrhein-Westfalen and Niedersachsen

5    of Germany. Figure 4(g) and Figure 4(h) show poultry emission is higher in southeast of the Netherlands but lower in Niedersachsen in the updated modeled results, but to a lesser extent. It can be seen from Figure 4(i) and Figure 4(j) that fertilizer application emission occupies only a small portion of the annual totals. The patterns are quite similar, except that the emission in northern France is lower in the new result. Besides, the mineral fertilizer emission estimates from MACC-III sometimes show higher values at country borders, which is not seen in MACC-INTEGRATOR. Because they use different allocation

10   methods: the original inventory uses proxy maps and emission fractions for different counties, while the updated one uses a balanced N fertilization approach at NCU level.





**Figure 4 Maps of annual emission total (Gg/yr) for all agricultural categories and for cattle, pig, poultry and mineral fertilizer in the year of 2010. The left panel indicates the results from the original MACC-III inventory while the right panel represents the output of the updated inventory. (a, b) emission from all agricultural sectors; (c, d) emission from cattle; (e, f) emission from pig; (g, h) emission from poultry; (i, j) emission from mineral fertilizer.**





## 3.2 Observed and modeled ammonia total columns

After filtering IASI ammonia total column product ANNI-NH3-v2.2R-I, the number of valid daytime overpass measurements in each month is illustrated in Fig. G1(a). The month in which the most valid measurements (more than 6000) occurred is April, followed by July and June in which there are nearly 3764 and 4862 measurements, respectively. The measurements in

these three months represent 74.4% of the total number over the whole year. Figure G1(b) shows the spatial distribution of measurement counts over the area of interest. Ammonia is measured validly mostly in Western Germany, Southern Germany bordering Austria, the Netherlands, Belgium and Northern France. The influence of satellite footprint on the availability of data leads to the strips which are more visible in Germany and France. Very few valid measurements are available over the sea and in mountainous regions.

The characteristics of valid measurements distribution correspond with area-averaged relative error (Figure 5a): the regions with few measurements tend to have a high relative error, while the lowest errors are witnessed in the Netherlands and Western Germany where lots of remote sensing data are available. For the rest of the area, the relative error is between 50% and 70 %. Figure 5b represents annual area-averaged ammonia total columns after post-filtering which excludes gird cells that have less than ten measurements, and an averaged relative error larger than 75%. In this way, the outcome is supposed to be more

reliable, and as a result, the effect of satellite overpass is more visually apparent. In regions with almost no agricultural activities, there are persistent low background columns around $0.8 \times 10^{16} \ molecules/cm^2$, because IASI is not very sensitive to low-level observations with low HRIs, which results in an overestimation of the observed columns.

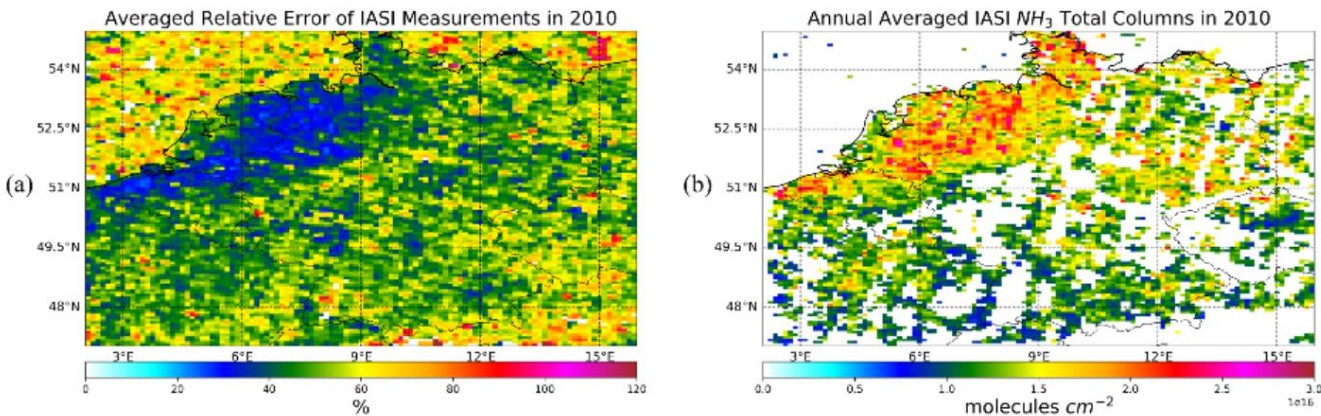

**Figure 5 The left panel (a) is the map of area-averaged relative error of IASI daytime measurements in 2010. The right panel (b) is the map of area-averaged total columns after filtering out grid cells that have less than ten valid measurements and an averaged relative error larger than 75%.**

The results of average annual total columns from LOTOS-EUROS simulations are shown in Figure 6, which are calculated

from 3-dimensional concentration outputs that are spatially and temporally closest to the satellite measurements overpass.





Overall, the new modeled result using updated annual emission distribution and time profile in Figure 6(b) gives a higher magnitude of ammonia columns than the original one. Bigger differences that are more than 100% increase occur mostly over Germany, while smaller changes of 0-50% rise are seen in the Netherlands and Belgium. The hot spots in the original simulations in Eastern Netherlands and the states of Nordrhein-Westfalen and Niedersachsen are more spread out and standing

out in the new simulations. Moreover, new hotspots appear in other regions such as southern Bayern and Baden-Württemberg close to the border with Austria and Switzerland.

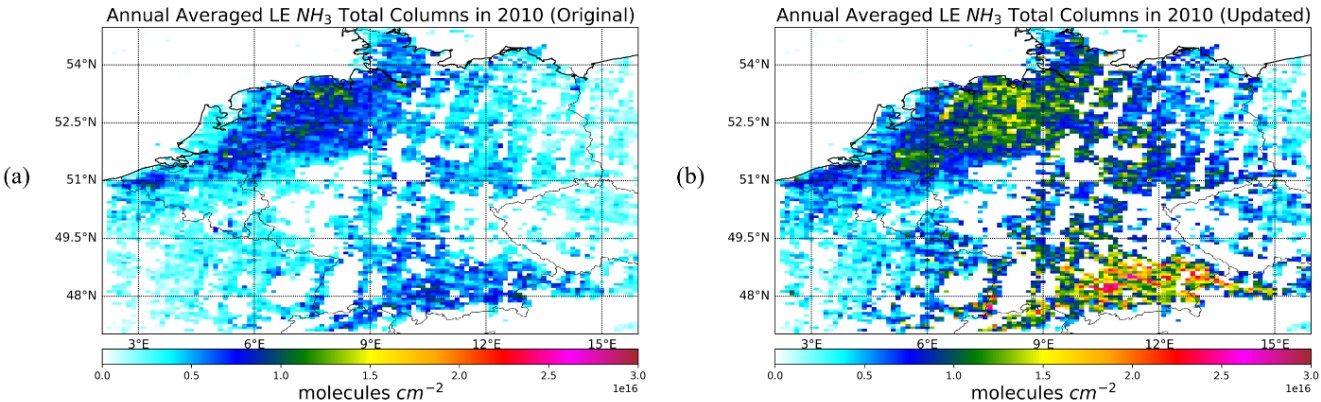

**Figure 6 The left panel demonstrates the simulated annual averaged total columns from LOTOS-EUROS using the original**

**MACC-III annual emission distribution and static time profile. The right panel shows the simulated annual averaged total columns from LOTOS-EUROS using MACC-INTEGRATOR emission totals and updated time profiles.**

Figure 7 shows scatter plots of comparisons between IASI observations and LOTOS-EUROS column estimates, with the left panel comparing the measurements with the original modeled result and the right panel comparing the measurements with the

updated output. Figure 7(a) and Figure 7(b) include all valid grid cells in Germany and Benelux. While simulated total columns could reach towards $0\ molec/cm^2$, IASI measurement has a minimum value of approximately $0.8 \times 10^{16} molec/cm^2$, which validates our observation from Figure 5(b). The simulated total columns from the original model are underestimated, while there exist both overestimation and underestimation in the updated modeled output. However, one can see in Figure 7(b) that there seem to be two plumes, one laying on the upper side of y=x and the other one laying on the lower side. The latitude of

each grid cell is indicated subsequently, and the two plumes are related to the latitude of the points within. For a clearer view, comparisons of totals column estimates and measurements on grid cells at lower latitudes (smaller than 49 degrees and located in Germany) are demonstrated in Figure 7(c) and Figure 7(d). The latter obviously illustrates that there is a considerable level of discorrelation in the south in the updated model, with measurements ranging between $1.0 - 1.5\ molec/cm^2$ while simulations varying from $0.5\ molec/cm^2$ to $3.0\ molec/cm^2$. This phenomenon is much less visible in Figure 7(c) because

the absolute values of the estimated total columns in the south are much lower, resulting in the less spreading-out distribution in the y-axis. These grid cells in the south will undoubtedly affect the performance during linear regression. Subsequently,





Figure 7(e) and (f) focus on the rest of the grid cells which are located at higher latitudes. Weighted linear regression was performed, with weight being inversely proportional to the square of the averaged relative error. From Figure 7(b) to Figure 7(f), the outcome of the new simulation has improved a lot. If purely based on linear regression output, the original output (Figure 7(f)) has a better slope but worse correlation than the original one Figure 7(e). Nonetheless, both perform rather poorly

5   if judged sorely by linear regression statistics and tend to underestimate total columns in the north.





**Figure 7 Scatter plots comparing ammonia annual averaged total column from IASI measurements and from LOTOS-EUROS model. The color of the points indicates latitude. The left panels and right panels use original and updated modeled results, respectively. (a) and (b) include all valid grid cell. (c) and (d) look at grid cell with lower latitude (< 49 degrees) while (e) and (f) focus on points with latitudes larger than 49 degrees.**



The performances of the original and updated model comparing with IASI observations were investigated based on four indicators: NRMSE, NMAE, EF and d. These indicators were computed for all grid cells within Germany and Benelux, as well as separately for each country (Table 2). Every indicator has improved for the new modeled results. Both NRMSE and NMAE have dropped, with the largest deductions of NRMSE and NMAE from Luxemburg and Germany. As for model

efficiency, even though the new modeled output gives values closer to one, they are still negative. Also, index of agreement witnessed the largest increase in the Netherlands and Luxemburg.

**Table 2 Performance assessment of the original and the updated model using NRMSE, NMAE, EF and d.**

|  | NRMSE | | NMAE | | EF | | d | |
|---|---|---|---|---|---|---|---|---|
|  | Original | Updated | Original | Updated | Original | Updated | Original | Updated |
| **All** | 35.51 | 27.00 | 65.95 | 46.83 | -7.85 | -4.12 | 0.35 | 0.38 |
| **DEU** | 41.34 | 30.26 | 65.28 | 43.98 | -7.94 | -3.79 | 0.35 | 0.37 |
| **NLD** | 44.00 | 35.95 | 66.06 | 53.02 | -9.61 | -6.09 | 0.35 | 0.41 |
| **BEL** | 66.00 | 58.20 | 70.84 | 61.66 | -10.96 | -8.30 | 0.33 | 0.36 |
| **LUX** | 100.15 | 76.61 | 78.79 | 59.66 | -17.71 | -9.94 | 0.27 | 0.34 |

Reichert (2016) showed that southern Germany is one of the areas in the country which has the highest density of cattle and pig livestock. Emission from housing, storage, grazing, and manure application are derived based on animal numbers, animal excretion, land use, fertilizer types, etc. at NCU level in INTEGRATOR. As a result, the emission from a certain animal, including housing, manure storage or manure application, occurs where the animal is located, without accounting for manure transport of regions with excessive manure to those with shortages. The role of manure transport is more significant when

there is a lot of animal livestock, such as Flanders in Belgium and the Netherlands where they reflect about one-third of the total applied manure (Hendriks et al., 2016). INTEGRATOR neglects manure transport, which could contribute to an overestimation in the south of Germany. Another factor that could cause the underestimation in the north and overestimation in the south is the emission fractions which describe the linear correlation between emission and excretion in housing/storage or amount of manure/fertilizer applied on crops. The emission fractions used in INTEGRATOR are only country dependent,

but they could vary from region to region because different regions have variabilities in feed, techniques, local climate, environment, etc., which undoubtedly should also be taken into account.

The feasibility of verifying emission estimates by comparing weekly or monthly time series derived from measurements and simulations was also investigated. However, since we only focus on the year of 2010 in this paper, the number of valid measurements is not sufficient for most grid cells to obtain reliable results. Moreover, as is illustrated in Fig. G1(a), the majority

of valid data are located in April, June and July, which makes it even more difficult to obtain a continuous time series. Consequently, two alternatives could be considered to resolve this issue. First, several years averaging is required for a better yearly time series. It is also possible to look at a longer time frame with coarser temporal resolution.





### 3.3 Observed and modeled ammonia surface concentrations

Figure 8 provides the scatter plots between paired in-situ measurements and LOTOS-EUROS simulations, showing all weekly or monthly averaged measurements (the temporal resolution depends on the measuring interval of the ground station). The updated linear regression result is better than the original one, with a slope closer to one and higher R-squared value. It also

5 appears that using the updated emission model yields a more coherent estimate with reality than the original model. Through coloring in Figure 8, the mid-day of the sampling period is indicated. In Figure 8(a), most of the blue points lie in the upper side of the fitted line and y=x, which tells that the original model usually overestimates surface concentrations (emissions) in the first 2 to 3 months of the year. In the meantime, the points in Figure 8(b) are more evenly distributed on both sides of the fitted line. If the scatter points in the first three months are excluded, as is shown in Figure 8(c) and Figure 8(d), R-squared

10 value does improve for the original modeled output, but the slope of linear regression is also worsened dramatically. On the contrary, filtering out measurements in the beginning months does not have an impact on the comparison between the new modeled results and measurements, both slope and R-squared almost remain the same, which implies that the performance of the updated model is more robust and stable.



**Figure 8 Scatter plots comparing ammonia surface concentrations from in-situ measurements and the LOTOS-EUROS model. The color of the points indicates time (day of a year). The left panels and right panels use original and new modeled results, respectively. (a) and (b) include all measurements and correspondent simulation results, while (c) and (d) exclude the data from the first three months of the year.**

Once again, the four indicators and correlation coefficient were calculated to determine the performance of the original and updated model (Table 3). All indices illustrate that the updated model has improved surface concentration estimates. The improvement in the Netherlands is much larger than that in Germany. The reason might be that the setup of ground stations is





more consistent in the Netherlands, namely the locations of the Dutch stations in the nature areas make them more representative for the overall emission temporal variation of a grid cell.

**Table 3 Performance assessment of the original and the updated model by comparing ammonia weekly (monthly) surface**
**concentration. Correlation, NRMSE, NMAE, EF and d are calculated using in-situ measurements and modeled results.**

|  | Correlation | | NRMSE | | NMAE | | EF | | d | |
|---|---|---|---|---|---|---|---|---|---|---|
|  | Original | Updated | Original | Updated | Original | Updated | Original | Updated | Original | Updated |
| **All** | 0.46 | 0.59 | 7.47 | 6.29 | 57.62 | 48.25 | 0.00 | 0.29 | 0.65 | 0.75 |
| **NLD** | 0.41 | 0.57 | 12.39 | 9.96 | 56.10 | 45.60 | -0.16 | 0.25 | 0.63 | 0.74 |
| **DEU** | 0.44 | 0.48 | 6.85 | 6.73 | 67.61 | 65.62 | 0.16 | 0.18 | 0.57 | 0.63 |

Figure 9(a) and Figure 9(b) shows the change in modeled surface concentration time series for the Station DEUB028 in Zingst, Mecklenburg-Vorpommern, Germany. The station is located in an agriculturally active region with cereals, industrial crops and animal housing. As can been seen in Figure 9(a), the original model does not correspond with the measurements well.
There is almost no ammonia measured before Julian Day 64, while the original model estimates there are two peaks on Day 38 and 59. Besides, the first two peaks in the measurement on Julian Day 80 and 110 are not captured by the original model. The updated model manages to simulate these two peaks, even though they are slightly delayed by ten days. The first and largest of the two peaks in spring is mainly explained by cattle manure application, followed by pig and poultry manure application, while mineral fertilizer contributes to a lesser extent. In the summer between Day 150 and 275, the new modeled
result also does a good job distributing ammonia emission temporally, with animal houses, cattle storage and mineral fertilizer application dominating ammonia emission.

A similar situation applies to station DEUB005 in Lüder-Langenbrügge as shown in Figure 9(c) and Figure 9(d), representing the validation of the original and updated model, respectively. We can see from Figure 9(c) that the original model again allocates substantial emission at the beginning of the year. The updated model improves the estimates a lot, even merged peaks
from spring mineral fertilizer and manure application are detected. However, there still exist two issues. One is that the peaks in spring between Day 64 and 140 are overestimated, the other one is that the whole time series seems shifted.

A possible reason for the delay of emission from fertilization is that the reference temperature sum in TIMELINES to estimate fertilization day is too large at this location, which will lead to a later predicted sowing day than reality. Another reason is likely the inclusion of impact from precipitation. The threshold of De Martonne-Index (1.7) could be too low at this location,
some days in January and February are considered to have excessive rain, so the whole curve is shifted to the right direction of the x-axis. Further improvement of the De Martonne-Index algorithm is in need since we extend the algorithm for Flanders to the whole area of interest, using the same threshold of the De Martonne-Index without considering regional differences. Moreover, Kranenburg used visual inspection to determine the threshold (1.7), more studies about De Martonne-Index should be done to correlate excessive precipitation and its impact on agricultural practices.

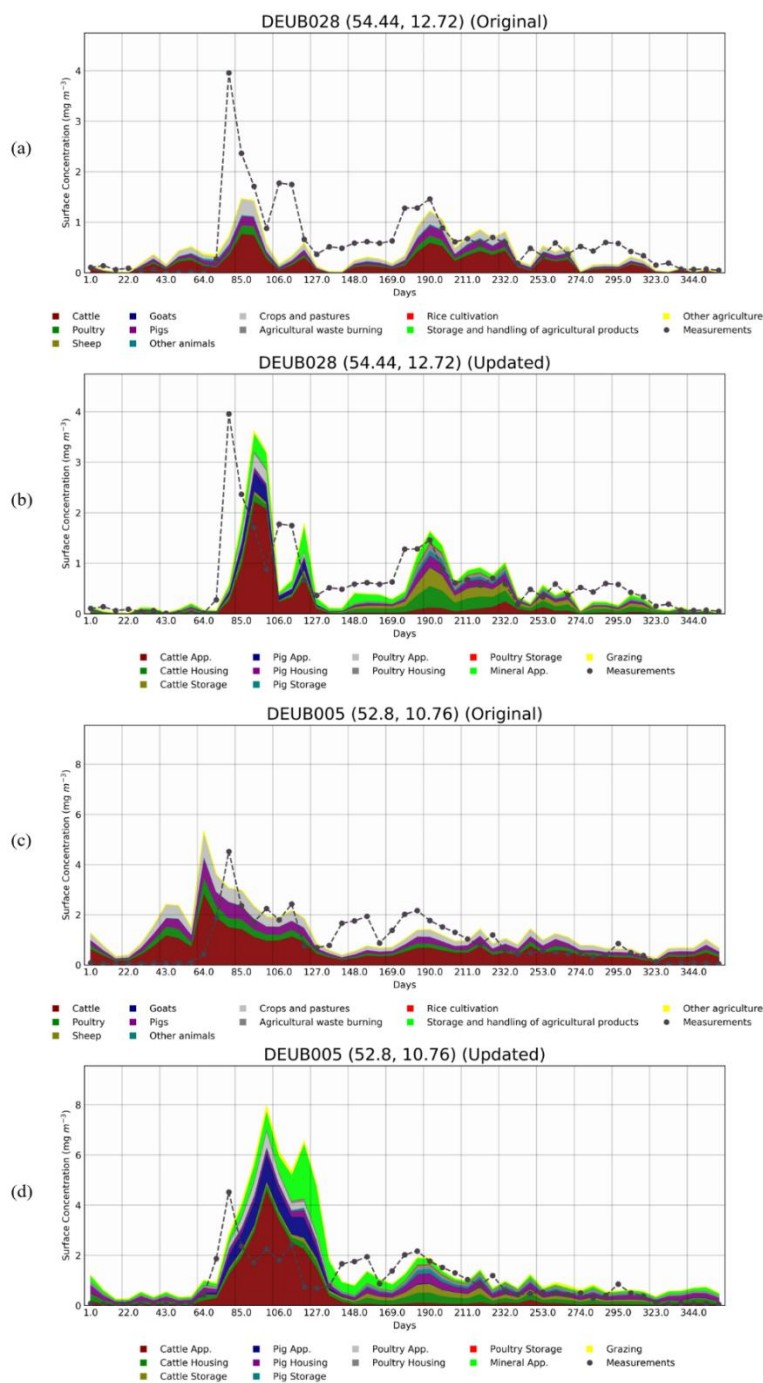

**Figure 9 Comparison of surface concentration measurements within EMEP network and simulated surface concentrations from original and updated modeled annual emission and time profiles: (a) in-situ measurements vs. original modeled output at station DEUB028; (b) in-situ measurements vs. new modeled output at station DEUB028; (c) in-situ measurements vs. original modeled output at station DEUB005; (d) in-situ measurements vs. new modeled output at station DEUB005.**





Another station in the region of Hanover, Lower Saxony is demonstrated in Figure 10. The measurements at this station only have a monthly temporal resolution. The updated model has shown much better correspondence with measurements than the original one, except that the average surface concentration in May is almost 50 percent higher. Figure 10(b) is able to point out that most of the agricultural activity at this location is related to manure and mineral fertilizer application, among which
5  cattle and pig manure application has the dominance. The overestimation in spring is thus most probably linked to overestimated emission from cattle or pig manure application. There are two possible contributions to this behavior. One is the emission fractions used in INTEGRATOR. The model uses country dependent emission fractions, which have been updated and detailed through others' studies that account for fertilizer type, climatology and soil properties, etc. Another reason is the way of resampling emission from NCU polygons to standard grids in LOTOS-EUROS. The emission with a polygon is
10  averaged all over the polygon evenly, which leads to misallocation. Last but not least, Lower Saxony is one of the states in Germany which has the highest density of livestock in the country. INTEGRATOR model calculates ammonia emission based on proxy maps of animal number and excretion input, without considering the fact that manure from this high production region could be transported to other regions where manure is in demand. This will also lead to an overestimation.

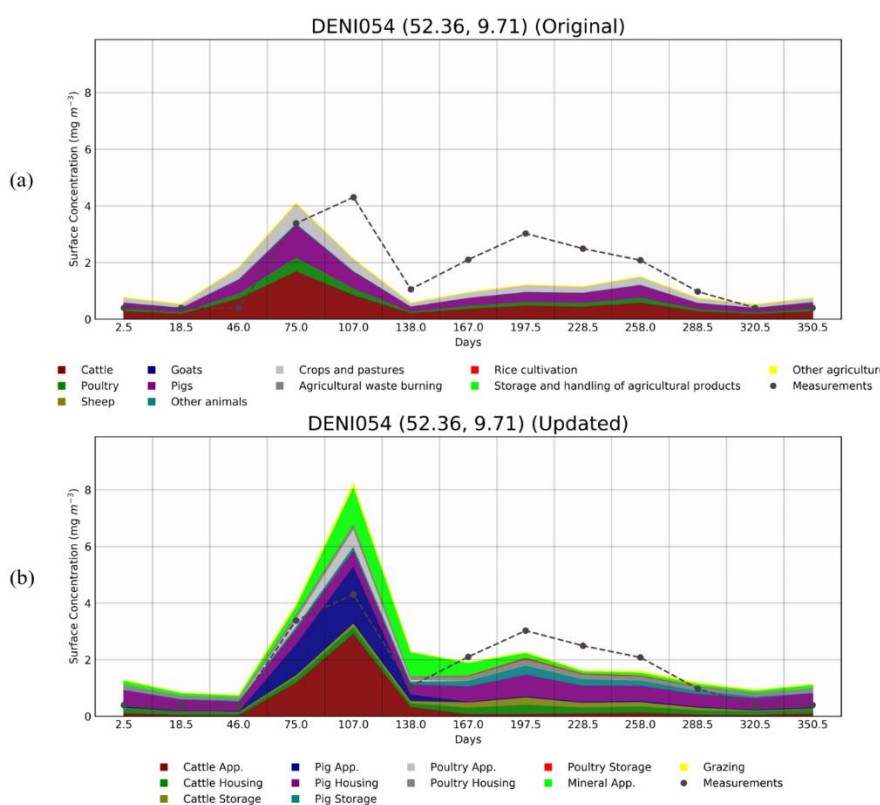

**Figure 10 Comparison of surface concentration measurements within EMEP network and simulated surface concentrations from original and updated modeled annual emission and time profiles at station DENI054. (a) in-situ measurements vs. original modeled output; (b) in-situ measurements vs. new modeled output.**



Figure 11 compares averaged surface concentrations over the year from the original and the updated model, for all ground stations as well as station located in Germany or the Netherlands. In all these three scenarios, the updated model shows improvement in the result of linear regression, regarding slope and R-value, with the larger improvement in the comparison

5   for Dutch stations, yet not very significant. In conclusion, the model is very useful to investigate the temporal variation and trend of ammonia emission and surface concentration and it does much better work than the original model. However, the improvement is not sufficient enough in the absolute sense, to derived reliable results of spatial distribution and characteristics of the annual average.







**Figure 11 Scatter plots comparing the annual averaged ammonia surface concentrations from all German and Dutch stations with the simulated annual average of the grid cells in which the stations are located. The left panels and right panels use original and new modeled results, respectively. (a) and (b) include all measurements and correspondent simulation results. (c) and (d) only take into account the measurements taken by German stations, while (e) and (f) only look at the comparison on Dutch sites.**



## 4 Discussion and Conclusions

*Impacts of manure and field characteristics and weather conditions on ammonia emissions*

The largest contributor to the difference in the measured and modeled spatial distribution of emission is from cattle (cattle housing, cattle manure application and storage, as well as grazing). A possible explanation of the difference is that the emissions fractions used in the INTEGRATOR are nation-wide averages, not accounting for impacts of application method, manure and field characteristics, and weather conditions. Those impacts have been studied by Huijsmans for both arable land (2003) and grassland (2001). He defined the formulation to describe the relationship between ammonia volatilization rate and the method of application and incorporation, the ATAN content of the manure, the manure application rate, the wind speed and the ambient temperature. Additionally, the empirical modeling of the emission process is carried out by RIVM and WUR using Volt'air approach (Holterman et al., 2014). Preliminary results show that the variations of weather conditions over the past 20 years lead to different emission fractions per month, and soil and manure characteristics also influence emission fraction. As a result, emission fraction differs at farm scale, contributing to inhomogeneous emission fraction on a regional or national scale. Moreover, as is mentioned in Sect. 2.2.3 The MACC-INTEGRATOR Combined , an empirical solution to keep the spatial pattern of INTEGRATOR emission distribution and apply scalars that were obtained per category at country level. Thus, the fundamental reason for an inaccurate emission allocation could be that local agricultural activity such as animal numbers and excretion inputs are inaccurate. These data are more accessible in countries like the Netherlands, Denmark, and Portugal. Therefore, there will be two steps for improvement regarding agricultural activity data and emission fraction. In the short term, we will implement detailed local activity data and update regional emission fractions for the Netherlands, Denmark, and Portugal, and investigate the difference brought by the refinement of the input. Next, a meta-analysis will be performed for parameterization based on local climatology, soil property, fertilizer type and application method to obtain emission fractions for other regions.

*Spatial detail of modeling and measurement locations*

The comparison of annual averaged surface concentrations at Dutch and German sites shows that the improvement in spatial details is not so significant. This may be caused by the restricted spatial representativity of measurement locations in combinations with substantial differences between sites within an NCU or a grid cell. During the resampling of emissions from NCU level to grid cells in latitude and longitude, emission estimates within an NCU from INTEGRATOR are evenly distributed all over the polygon, regardless of the actual locations of crops, animal houses, and manure storage facilities. Hence, for some locations where NCU polygons are relatively larger in size, spatial characteristics such as hot spots will be smoothened out to the whole polygon. In addition, some NCUs are composed of multiple disconnected polygons, within only some of which a certain crop, animal house or manure storage is present. The current algorithm will assign emission to other polygons sharing the same NCU number. High-resolution crop maps could help locate emission from fertilization on cropland/grassland inside polygons, as well as verify crop type predictions in the INTEGRATOR model. Inglada et al. (2015) assessed the state-of-the-art supervised classification methods and produced more accurate crop type maps with high resolution



multi-temporal optical imagery from SPOT4 (Take5) and Landsat 8. Surface reflectance, NDVI, NDWI and brightness were chosen as features, random forests and support vector machines (SVM) were selected as classifiers. Belgiu and Csillik (2018) proposed a time-weighted dynamic time warping (TWDTW) method that uses NDVI time series obtained by Sentinel-2 data for classification. It was proved to be more efficient in terms of computational time and less sensitive in relation to the training

samples, which is important for regions where inputs for training samples are limited. Besides Sentinel-2 optical images, Giordano et al. (2018) also included Sentinel-1 radar measurements to help crop classification using the complementarity between the multi-modal images, where Sentinel-2 may suffer from cover while Sentinel-1 radar images allow getting more information. We will use make use of the above methods and more as well as available training samples (actual land use) in the Netherlands and Germany to obtain crop maps with high spatial resolution and allocate emission from manure and fertilizer

application in a more precise way.

*Temporal detail of field operations*

Regarding the newly developed temporal allocator, we made modifications and updates in the parameterization proposed by Skjøth (2004, 2011) and Gyldenkærne (2005) who developed a dynamical ammonia emission parameterization which accounts for the agricultural activities and differences, based on the meteorological variables wind speed and surface temperature, as

well as the ventilation and heating inside stables. The first modification is that subsectors of emission from manure/fertilizer application were created to adapt to the emission sectors in INTEGRATOR. The corresponding emission functions were obtained by replacing $\mu$ in Eq. (1) with emission peak day obtained by estimated fertilization day from the TIMELINES model, while other parameters such as temperature and $\sigma$ remain the same. Agricultural models, including TIMELINES, usually work from the perspective of maximizing the efficiency of nitrogen use. However, farmers are likely to choose to apply them when

labor and machinery are both available and are unlikely to finish manure application in one day on the farmlands with the size of an NCU, leading to the inaccuracy between fertilization day estimate and reality and an extended manure application period. Moreover, the TIMELINES model heavily depends on the empirical data on sowing and harvesting dates currently used within CGMS to calculate the thermal time thresholds. The data are out of date and limited regarding the variety of crops, making it capable of simulating the timing of field operations for some but not all arable crops at different locations across Europe.

Consequently, a more thorough analysis is needed to refine the relationships between different field operations (Hutchings et al., 2012). Besides, soil moisture, workability and trafficability might improve the prediction of plowing and applications of solid manure in preparation for spring crops are made in the previous autumn. Another change to Gyldenkærne (2005)'s method is the implementation of legislative constraints and the impact of excessive precipitation. Fertilizer and manure application are not allowed on Sundays and outside permitted fertilization dates, which improved the emission estimates

outside the growing season and during winter by restricting emission application within a certain period (different per country), while the original model usually mistakenly allocates emission at the beginning the year since it uses simplified and static time profiles for all agricultural categories.

*Quality of total column satellite data*



The quality of total column simulations relies on the assumption that the spatial and temporal distribution of the emissions in the LOTOS-EUROS model closely represent reality. As is mentioned previously, detailed crop maps can help allocate emission from manure and fertilizer application within polygons. What is more, because only modeled ammonia columns at overpass time are selected for averaging, the accuracy of the seasonal variation in the $NH_3$ emissions in LOTOS-EUROS is therefore

of great importance. Dammers et al. (2016) found that the validity of the IASI product is quite limited because the satellite retrievals are biased. The retrieval of ammonia columns from IASI is still an on-going process, with a few studies having examined the quality of the products. Further development and validation of the IASI retrieval are very much in need for the understanding of the satellite's product. It remains poorly validated with only a few dedicated campaigns performed with limited spatial, vertical or temporal coverage. The key finding of the previous studies on the retrieval is that vertical profiles

of ammonia distribution has lots of uncertainties and need to be improved. Dammers et al. (2016) suggested that tower measurement campaigns are very important and helpful towards a better understanding of the vertical profile. Li et al. (2017) showed that there is a clear seasonal variation in the vertical distribution of $NH_3$ and that the slope of the $NH_3$ concentration gradient varies throughout the year, observing relatively high $NH_3$ ground concentrations during winter. His reasoning was that boundary layer is shallower in winter, which will potentially trap ammonia emissions and reduce $NH_3$ concentrations higher

up the column. As a result, IASI could miss high $NH_3$ ground concentrations in winter because of the lack of sensitivity to the lower parts of the boundary layer. On the contrary, most of the valid measurements used in this paper to calculated annual average are in April, June and July in which weather is relatively warmer. The boundary layer is consequently thicker, especially during clear-sky daytime condition in which IASI observations are utilized. $NH_3$ concentrations could be overestimated in higher altitude because it is more sensitive to the upper parts of the boundary layer. Recently, new products

become available, making it possible to cross-check results among satellites. CrIS is one of the new products that deserve attention, having the advantage of acquiring more explicit information on the sensitivity of the satellite (averaging kernel). Another uncertainty comes from possible manure transport. INTEGRATOR assumes that the emission from a certain animal, including housing, storage or manure application, occurs where the animal is located, ignoring manure transport of regions with excessive manure to those with shortages. Hendriks et al. (2016) looked into manure transport data in Flanders and found

that the manure transport data account for roughly 1/3 of the amount of manure used in Flanders each year, while the remaining 2/3 consists of manure that farmers apply on their own land. The pattern of manure transport can be used as a proxy for the temporal pattern of ammonia emissions from manure application, under the assumption that manure is applied to the fields on the day of transport.

*Quality of in-situ ammonia surface concentration measurements*

The time series of surface concentrations from simulations and in-situ measurements show better alliance than annual averaged total columns, making it possible to visually detect the ammonia brought by various agricultural activities. There could be an occurrence of inconstancy. First, sometimes overall magnitude is not in accordance with measurement, because emission estimate of a grid might not be accurate during spatial allocation, which is related to the flaws of the spatial allocator mentioned above, namely crop allocation within NCU, out-of-date emission fraction, etc. Moreover, simulations could be shifted




horizontally compared to measurements, which is possibly caused by two factors. One is that fertilization day estimate needs correction, which is calculated by outdated reference thermal sum. The other reason could be the threshold of De Martonne-Index applied to the area of interest. Fertilization is also considered unfeasible during days with heavy precipitation which is determined using De Martonne-Index. However, the threshold was decided with a visual inspection for Flanders and expanded

to all regions. Sometimes the emission from application might be delayed when the model wrongly considers precipitation to be too excessive for fertilization. Further inspections are in demand to obtain a location-dependent threshold for a larger-scale area. Finally, sometimes the modeled time series completely mismatches measurements, on which the location of the ground station has a significant impact. In-situ measurements represent the ammonia emission characteristics of rather a point source. The spatial resolution of the updated model is around 7km by 7km which is relatively course compared to the spatial

characteristics of agricultural land and facilities, averaging and smoothening the spatial details of ammonia. A station next to animal houses or manure storage facilities will result in measurements of constant high level over the year. A station in a forest is also not appropriate for validation since the short atmospheric residence time of ammonia as it is effectively removed by dry and wet deposition several hours after emission. Most ideally, a station next to arable land but is distant from an animal house or manure storage would be most optimal in this paper to verify the timing of emission from manure/fertilizer application

obtained with the methodology of the TIMELINES model.

*Conclusions*

In summary, this paper is a new attempt to build an ammonia emission model which is composed of a spatial allocator that has a relatively high spatial resolution and can distinguish various agricultural sectors including crop types, fertilizer types, animal houses and manure storages, and a temporal allocator that is spatially explicit and dynamic based on land use, local climatology

and legislative constraints. The updated model overall estimates more emissions, with the country total being 14% higher in Germany, and 6.6% higher in Benelux. Extra new hot spots appear in southeastern Germany. Despite the limitations in modeling and data for validation, LOTOS-EUROS performed better with the updated emission product, especially in the representation of the temporal behavior of ammonia concentrations. Comparison between modeled and observed ammonia levels show much better correspondence and more robust performance when using the updated emission information,

especially the temporal variability is captured better as the new methodology successfully differentiates regional variability in seasonality in ammonia emissions. The distribution of annual emission obtained from the updated model is similar to that from the original MACC-III model. The labeling module of LOTOS-EUROS helps us trackback the emission sector of the modeled ammonia surface concentration and total columns for better interpretation and future improvement. When reliable detailed input datasets are available and the methodology is further improved as described, we can expect to extend this approach to

Europe.



## Appendix A

In order to assess the performance of the updated model and compare it with that of the original model, normalized root mean square error (NRMSE), normalized mean absolute error (NMAE), model efficiency (EF) and index of agreement were calculated.

5 The root mean square error of n predicted values of a regression's dependent variable, with $\hat{y}_i$ being the i-th prediction and $y_i$ being the i-th estimate, is computed as the square root of the mean of the squares of the deviations:

$$RMSE = \sqrt{\frac{\sum_{i=1}^{n}(\hat{y}_i - y_i)^2}{n}} \qquad (A1)$$

The NRMSE indicates RMSE in a relative sense, by dividing RMSE by the difference between the maximum and minimum observed values:

$$NRMSE = \frac{RMSE}{y_{max} - y_{min}} \qquad (A2)$$

Normalized mean Absolute Error (MAE) is interpreted as the average absolute difference between $y_i$ and $\hat{y}_i$, with reference to the mean of observations:

$$NMAE = \frac{\sum_{i=1}^{n}|\hat{y}_i - y_i|}{n} / \bar{y} \qquad (A3)$$

Model efficiency coefficient is used to illustrate predictive power. It can range from $-\infty$ to 1. An efficiency of 1 indicates a
15 perfect match of simulations to observations (Ritter and Muñoz-Carpena, 2013). The closer the model efficiency is to 1, the more accurate the model is.

$$EF = 1 - \frac{\sum_{i=1}^{n}(\hat{y}_i - y_i)^2}{\sum_{i=1}^{n}(y_i - \bar{y})^2} \qquad (A4)$$

In addition, an index of agreement (d) statistic was also employed, which represents the ratio of the mean square error and the potential error (Willmott, 1981). The agreement value of 1 indicates a perfect match, and 0 indicates no agreement at all.
20 However, it is overly sensitive to extreme values due to the squared differences (Willmott, 1981).

$$d = 1 - \frac{\sum_{i=1}^{n}(\hat{y}_i - y_i)^2}{\sum_{i=1}^{n}(|\hat{y}_i - \bar{y}| + |y_i - \bar{y}|)^2} \qquad (A5)$$



**Appendix B**

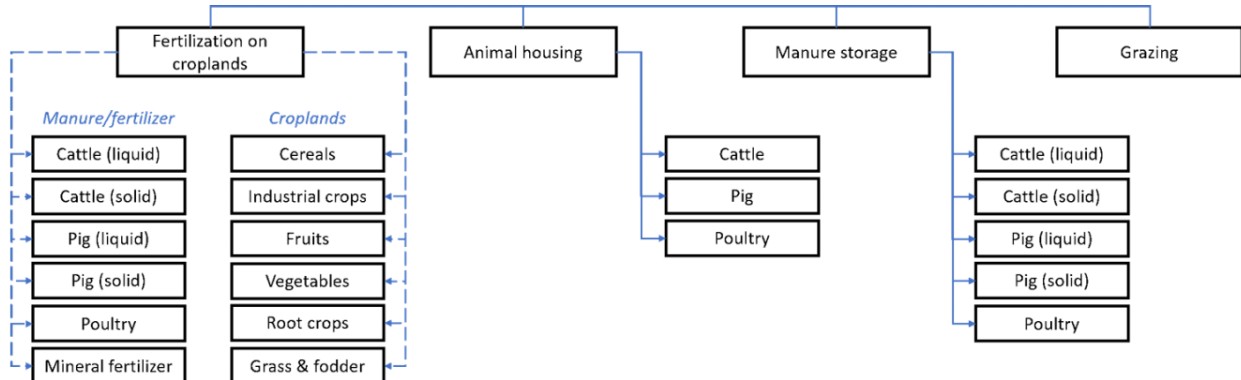

**Figure B1 Categorization in the MACC-INTEGRATOR combined emission inventory. There are 6 fertilizer type and 6 crop types, resulting in 36 categories regarding fertilization. Together with 3 animal housing types, 5 manure storage types, and grazing, there are 45 categories in the new ammonia emission model.**





**Appendix C**

A comparison between sowing and harvesting days calculated for this paper and by Hutchings et al. (2012) is made and shown in Fig. C2 for verification. Figure C2 depicts an example of the calculated sowing days of potatoes. Only the dates for years between 1985 and 2000 are selected for comparison because Hutchings et al. (2012) used predicted temperature data for years
5    after 2000. The sowing days generated are in good alignment with only a few outliers away from line y=x.

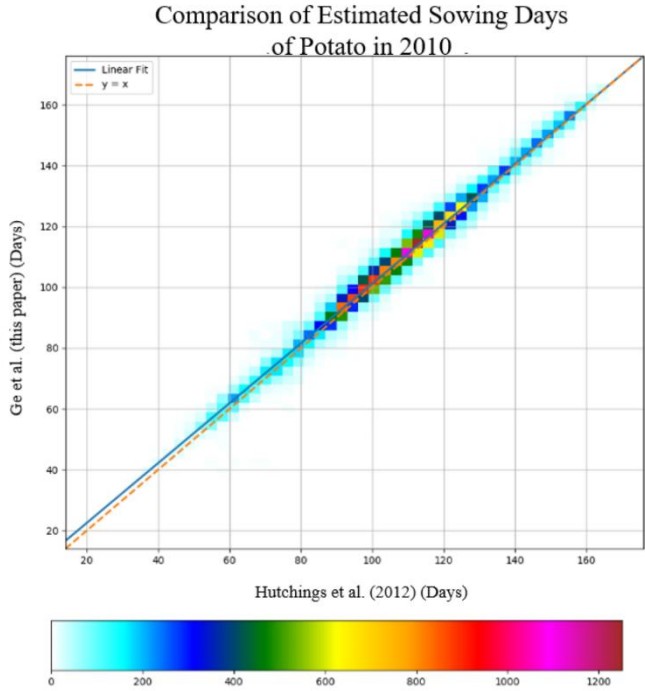

**Figure C1 The comparison of sowing day estimates between 1985 and 1995 by Hutchings and in this study. The left panel is the scatter plot of sowing days obtained from Hutchings (x-axis) versus those generated in this study (y-axis). The right panel is the density plot of the left panel, with the color indicating the number of points lying in a grid cell.**





**Appendix D**

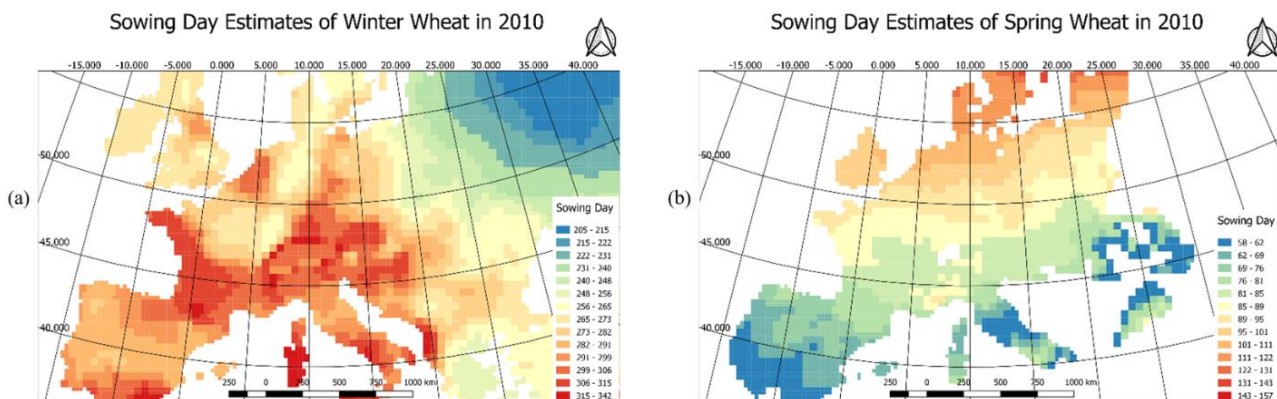

**Figure D1 Two examples of estimated sowing day over Europe from the TIMELINES model for winter wheat (left) and spring wheat (right) in the year 2010.**





## Appendix E

As is shown in Fig. E1, the weekly De Martonne-Index at location coordinate (48.98, 8.14) approximately ranges between 0 and 6.5 in 2010. High indices are observed around Day 30 before the first spring application period as well as at the end of the year. On these occasions, the index reaches values well above 3.

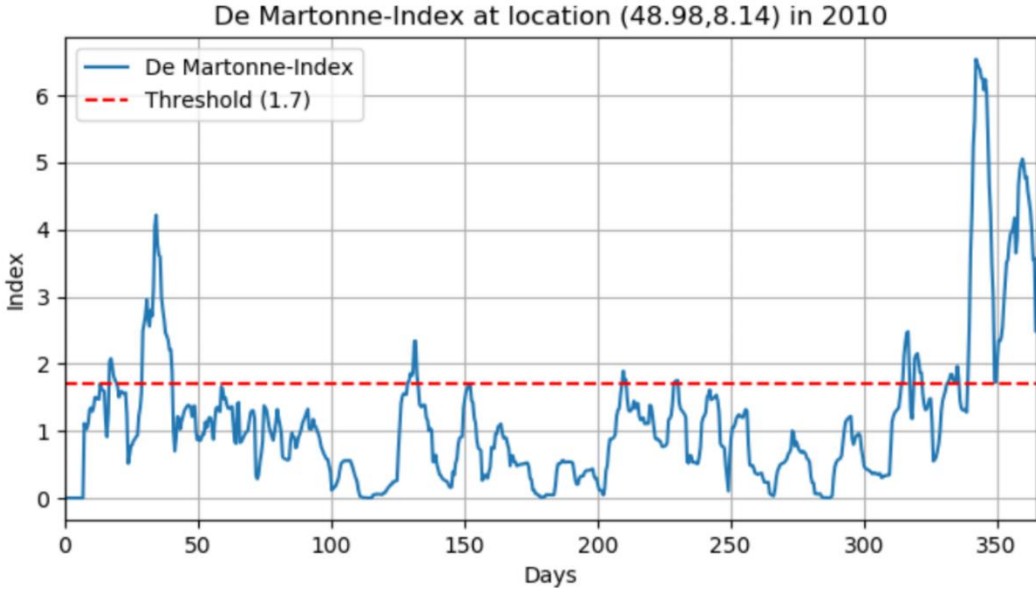

**Figure E1 An example of the time series of the weekly De Martonne-Index, at (48.98, 8.14). A threshold of 1.7 is determined, above which precipitation is considered to be excessive.**



**Appendix F**

**Table F1 Information on the selected in-situ measurement sites.**

| Station Code | Network | Latitude | Longitude | Existing Land Use |
|:---:|:---:|:---:|:---:|:---|
| DEUB028 | UBA | 54.44 | 12.72 | Cereal, industrial crop, grassland, manure storage, animal housing |
| DEUB005 | UBA | 52.8 | 10.76 | Cereal, root crop, industrial crop, grassland, manure storage, animal housing |
| DENI054 | UBA | 52.36 | 9.71 | Cereal, root crop, industrial crop, grassland, manure storage, animal housing |
| DEBY151 | UBA | 47.81 | 10.72 | Grassland, manure storage, animal housing |
| NL63-4 | MAN | 51.40 | 5.66 | grassland, manure storage, animal housing |





**Appendix G**

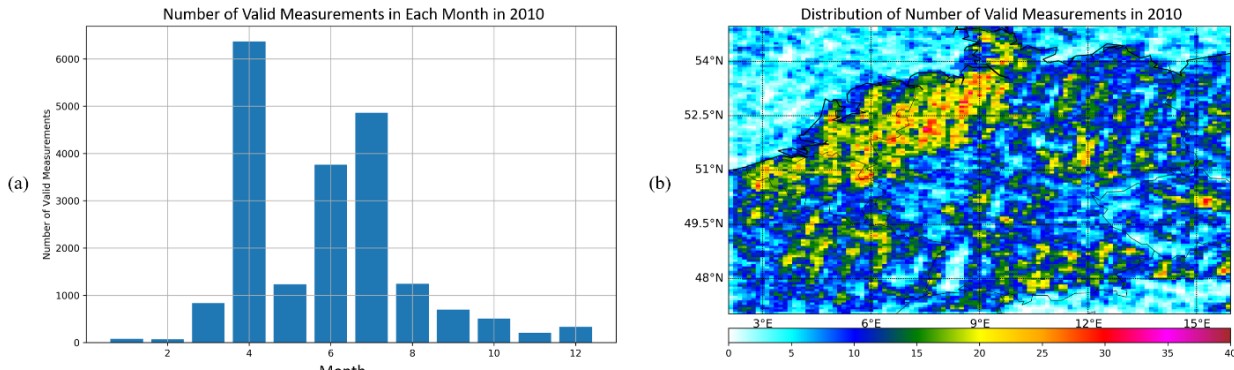

**Figure G1 Left panel (a) is the bar plot of the number of valid IASI measurements as function of measuring time (month). Valid measurements mean the relative error is less than 75%, cloud cover is smaller than 10% and the measured value is a positive number. Right panel (b) is the spatial distribution of number of valid IASI measurements.**



*Data availability*. The updated annual ammonia emission distribution and corresponding time profiles are available by request.

*Competing interests*. The authors declare that they have no conflict of interest.

5 *Author contribution*. Xinrui Ge designed and programmed the processing chain, performed the simulations and analyzed the results for discussion and conclusion. Martijn Schaap is the daily supervisor of the project and provided with his expertise in atmospheric modeling and sciences. Martijn Schaap, Richard Kranenburg and Arjo Segers designed the model code of LOTOS-EUROS. Gert Jan Reinds helped with the technical issues of TIMELINES and INTEGRATOR. Wim de Vries is the promotor of the project. He and Hans Kros offered their knowledge regarding nitrogen use and ammonia emission from
10 agriculture.

*Acknowledgements*. This work is a part of the AMARETTO project which is financially supported by NWO. We would like to thank Dr. N. Hutchings for providing the source code of the TIMELINES model and the reference sowing and harvesting dates data. We thank the reviewers for their constructive comments.





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
