# Peer review of "Modeling Atmospheric Ammonia using Agricultural Emissions with Improved Spatial Variability and Temporal Dynamics"

_Atmospheric Chemistry and Physics, 2019_

## Short Comment (SC1) · 27 Apr 2020

In this study, the authors develop a new system to estimate ammonia emission over Germany and Benelux. They compare their results to measurements such as the ones from the IASI mission. I have few advices in the use of these observations. In their manuscript, the authors mention they've used the ANNI-NH3-v2.2R-I products and they've filtered the data based on: - on cloud coverage (< 10%), - column values (positive), - relative absolute errors (<75%)

The data should already include only the observations with a cloud cover lower than 10%, so this filter is not necessary. In the use of the positive data, I also suggest using

the negative columns. It is not straightforward to do it, but by using only the positive columns, the authors will probably overestimate their averages. The use of these negative columns will help you to get a background close to 0. It is something already done for other trace gases, see for example with the IASI HCOOH data: https://www.atmos-chem-phys.net/16/8963/2016/ (last paragraph section 2.3.1). On the last filter, the authors should justify the 75% threshold. The larger errors given in the IASI data set are often related to the lower NH3 total columns and, as with the filter on positive columns, the authors take the risk to overestimate their IASI averaged distribution.

Thus, I suggest using the full data set. As currently done in the manuscript, the authors have only kept the large total columns which might explain the differences noticed in the scatterplots (e.g. Figs. 7a and b).

The best advice I can provide is to contact directly the authors of this IASI NH3 dataset if it was not done before: Martin VanDamme (mvdamme1@ulb.ac.be) and Lieven Clarisse (lclariss@ulb.ac.be).

---

## Author Comment (AC1) · 26 May 2020

Thanks for your feedback on our paper.

During the comparison of annual averaged columns, we filtered the measurements based on relative error (and other criteria) to ensure that we used observations with smaller uncertainties. We think you have a point that if we do so, the averaged columns will be most likely overestimated because smaller columns tend to have larger relative errors and to be excluded. As a matter of fact, filtering based on either relative error or absolute error will lead to biases in the outcomes.

Therefore, we adopted your suggestion and used all measurements (including negative columns) for validation. As can be seen in Figure 1, the background level in annual averaged columns has been eliminated. Subsequently, we calculated overpass modeled columns that are closest to measurements in space and time from the original and updated models. Compared to Figure 7(a) and (b) in the paper, the scatter plots in Figure 2 here show better linear correlation and less randomness between observations and simulations. However, the general characteristics remain the same, namely the updated model tends to overestimate ammonia in the south and underestimate in the north, while the original model has the tendency to underestimate regardless of latitude.

As a result, we can conclude that the use of all IASI data will greatly help to eliminate the background level of ammonia and improve the linear correlation between observed and model columns, but it does not have a large impact on what has been discussed in the paper. The plots here will be updated in the final version of the paper. Thanks for your time!

---

## Referee Comment (RC1) · Anonymous Referee #1 · 24 Jun 2020

General comments This remains a very long paper and some parts resemble more a good working draft written by a PhD student than a final draft that has had the guiding hand of an experienced scientist. In this respect, the authors are doing themselves a disfavour, since the work is otherwise something of a tour de force. In particular: - The English needs to be improved. I started making corrections to the language but stopped at page 24, as I was being distracted from my main task (considering the science). - Some of the text in the Results section is actually a discussion of the results and should be moved to the Discussion section.

As I noted in my original review, I am surprised that the authors did not choose to

compare their simulations with measurement sites that did not have local ammonia sources. Such sites would be expected to reflect the consequences of agricultural practices over a wider area.

Use of 'ammonia' and 'NH3' is inconsistent. I would suggest using NH3 (once it has been defined), except when it starts a sentence.

Specific comments

Page 7 What do the authors mean by 'non-fertilizer N input'? Is this biological N fixation or atmospheric deposition (or both)? Replace 'available manure' with 'available manure N'

Page 8 The authors write: It has to be noted that the ammonia emission estimates from INTEGRATOR differ from the officially reported national emission totals which are used in the MACC-III inventory. Because each country utilizes its own estimation algorithms that deviate from the INTIEGRTOR methodology which starts with animal number, excretion rate and emission fraction.

The national methodologies will also start with animal number and excretion rate, though they may not be the same as those used in INTEGRATOR. I suggest the following formulation: It should be noted that the NH3 emission estimates from IN-TEGRATOR differ from the officially reported national emission totals that are used in the MACC-III inventory. This is because each country uses its own emission inventory methodology whereas INTEGRTOR uses a uniform methodology for all countries. Change 'time profiles that distribute annual emission total in a grid cell over the course of a year' to 'time profiles that distribute the annual emission total in a grid cell over the course of a year'

Page 10 The authors write: Even though the difference between daily mean temperature and the base temperature is larger in the south, the greater reference thermal sum makes it longer to reach. Whereas for spring wheat, the reference thermal sum in the

south is less significantly bigger than that in the north, resulting in earlier sowing day in the south. I think this would be better: For winter what, even though the differences between daily mean temperature and the base temperature are larger in the south, the greater reference thermal sum it takes a longer time to reach this thermal sum. Whereas for spring wheat, the reference thermal sum in the south is less than that in the north, resulting in earlier sowing day than in the north.

The authors write: Also, Gyldenkærne et al. (2005) argued that there still expect to be variation in the timing of fertilizer and manure application because farmers often spend several days applying fertilizer and manure to the field. This means that a normal distribution around the central estimate and Gaussian functions are used to characterize it.

I think this would be better: In addition, Gyldenkærne et al. (2005) argued that there would still be variation in the timing of fertilizer and manure applications because of the time it would take farmers to complete these operations. As a consequence, a normal distribution around the central estimate was used here and Gaussian functions used to characterize it.

The authors write: Except that spring-summer application's deviation is 16 days, the other applications are given a deviation of 9 days. Besides, mineral fertilization in early spring and summer have a deviation of 9 and 16 days, respectively. Therefore, in this paper, we followed the systematic: for fertilizations that lie between mid-May and mid-August the deviation of corresponding emission function is 16 days, while for the others the standard deviation is considered to be nine days. I think this would be better: The standard deviation of the spring-summer application is 16 days while that of the remaining applications was 9 days. The standard deviation of the timing of the mineral fertilization applications in early spring and summer were 9 and 16 days, respectively. We make a similar assumption in this paper: for fertilizations that lie between mid-May and mid-August, the standard deviation of the corresponding emission function is 16 days, while for the remainder, the standard deviation is considered to be 9 days.

The authors write: intrusion effectiveness of manure and lead to an enormous amount of nutrient losses. Besides, after excessive precipitation, it is much more difficult for farmers to perform fertilization practices I think this would be better: infiltration rate of liquid manures and would risk serious surface runoff. In addition, trafficking the wet soil surface with heavy machinery would likely be impossible.

I am unclear what the authors mean by: Therefore, manure application is not effective during these conditions. As far as I can judge, this sentence is not necessary, since the subsequent sentences define how the model treats this situation.

Page 16 The abbreviation IASI is used before it has been defined: The following sentence does not make sense: Furthermore, the retrieval algorithm if IASI needs an accurate temperature profile, without which larger measurement errors will occur. I guess they mean: Furthermore, the retrieval algorithm of the IASI needs an accurate temperature profile, without which large measurement errors will occur.

The authors write: Even though for the comparison with modeled concentrations, the used measurement data should be representative of a wider region, it is impossible to get rid of local influences completely. In this study, all measurements were also looked to determine the overall performance of the original and updated model I think this would be better: For the comparison with modeled concentrations, the measurement data used should ideally be representative of a wider region. However, with measurements made at a single location, it is impossible to remove the local influences completely.

I do not understand this sentence: In this study, all measurements were also looked to determine the overall performance of the original and updated model

Page 17 The authors write: Because thermal contrast

I think this would be better: This is because the thermal contrast

Page 18 The authors write: Because they use different I think this would be better: This is because they use different...

Change caption above Fig 4a to read '... total NH3...'

Page 21 The authors write: the outcome is supposed to

Supposed by whom? Need a reference here.

Page 22 The authors write: standing out I think this would be better: prominent

Replace 'plumes' with 'clusters'. Replace 'demonstrated' with 'shown' Replace 'the less spreading-out distribution in the y-axis' with the lower dispersion along the y-axis'

Replace 'sorely' with 'solely'.

Replace 'look' with 'show' in Fig 7 caption The authors need to make the captions above the figures more meaningful. They are all identical at the moment, which means the reader has to read the main figure caption to identify what they show.

(End of English copyediting at page 24)

Much of the second and third paragraphs does not present the results but discuss them. This text needs to be moved to the Discussion.

Page 28 Again, much of the second and third paragraphs does not present the results but discuss them. This text needs to be moved to the Discussion.

ATAN is not defined.

'Preliminary results show' – preliminary results of what and by whom?

Page 34 The 6 to 7 lines from 'Regarding the newly developed temporal..' repeat the description given in the Methods section and can be removed.

The abbreviation 'CrIS' is not defined.

'Moreover, simulations could be shifted horizontally compared to measurements' I think they mean that the simulation could be systematically lagging in time behind the measurement.

The authors state that: Most ideally, a station next to arable land but is distant from an animal house or manure storage would be most optimal in this paper to verify the timing of emission from manure/fertilizer application

If one was assessing the ability of a model to simulate agricultural field operations, it would not be sufficient to do so at just a single site.

---

## Editor Comment (EC1) · Jayanarayanan Kuttippurath (Editor) · 20 Jul 2020

Review of

**Modeling Atmospheric Ammonia using Agricultural Emissions with Improved Spatial Variability and Temporal Dynamics, Ge, et al.**

The MS deals with modelling of ammonia emissions from agriculture and allied sector. They use a new model (a module) by incorporating the additional emission sources as compared to those in the existing models. This is a new attempt and can be considered for a publication in ACP. However, the article needs substantial revision before it can be accepted.

1. The MS is too long and was also a bit difficult to read because of the style of writing. Therefore, kindly consider shortening the length of the article and rephrase the sentences as suggested below, to the least. I have given only some examples. Please go through the entire article and check.
2. I thought one more year is needed to test the new model so that seasonal cycle can be tested and inter-annual differences can examined. If you have the additional model runs, please include and discuss.
3. There are a number of statements about IASI measurements. Perhaps, you could invite one of the IASI team members as an author. This would improve the content of the article. It is just a suggestion.
4. I find the improvement (the difference too) is mostly for the simulations for Germany, not for other countries. Is there any reason for this?
5. Use either "Deutschland" or "Germany" throughout the article, but do not mix both.
6. Other specific comments are given below and please attend them carefully.

**Page 1**

Line 12: unavailability of measurements?  Many satellite-based measurements are available (TES, IASI, CrIS, AIRS, and TANSO-FTS)

Line 22: Benelux means all three countries, then clearly state Belgium, Netherlands and Luxemburg

Line 24-26: difficult to understand, please rephrase the sentence

Line 30: "and model simulations"

Line 31: What about other European countries?

Line 29-32: Please state the reason for the differences.

**Page 2**

Line 9: I thought biomass burning contributes significantly (13-16%).

Line 17: "they" means PM?

Line 19: what is radiance balance?

**Page 3**

Line 20: "Level 1 Category"

Line 21: "application of the model results"

**Page 4:**

Line 6-9: Then we evaluate the model results by comparing the simulated total column and surface concentrations of ammonia with ..

Line 8-10: Finally, we evaluate the model performance with respect to improvements and shortcomings of the modelled results ..

**Page 5**

Line 1: this is a repetition of an earlier sentence

Line 5, 7: Two model runs were performed

Figure 2: please write "application"

**Page 7:**  Line 2: What is "expert-based judgement" here?

**Page 8**

Line 2: Please state how much ammonia emits from the traffic sector (in %)?

Page 10: Line 26-29: please rephrase, I did not understand this sentence.

Page 11, Line 1: You mean, no previous research studies?

Line 14: "from to"  delete "to"

Page 12, Line 4-5: How much impact that would make on the results, if you drop the constant for simplification?

Page 15: Line 1, what is "high lower critical temperature"

Page 15: Line 13, "showed a good agreement "

Page 16: Line 8: What does it mean by, "but it is not sufficient for real time monitoring?"

Page 16, line 9-10; Instead of this sentence, you can write the accuracy of the satellite product. This is a validated satellite instrument. Alternatively, you can give the details in Section 2.5.2

Page 17, line 8: If the uncertainty is 1000%, how can we use/trust the data or results? I thought the uncertainty is in the range of 25-50%, depending on the region. Perhaps, you need to check the IASI validation papers again.

Line 9-11: Yes, I agree. Thermal contrast might induce some uncertainty. However, the IASI validation team has recommended only the daytime measurements for scientific analyses. Please refer the IASI validation papers.

Line 18: weighted mean or weighted average

Line 31: you mean "poor classification of emissions in the MACC"?

Page 18: Line 6, "increases" instead of ascends

Line 7: scaling "we applied"

**Table 1**: Can we say that the emissions updated are not very different from the original, except in Germany? Is there any reason for this?

**Figure 4**: Too difficult to see the x and y - axis entries (please enlarge the font size). Please consider improving the other figures too.

Page 21: Line 1: please write just the "IASI measurements" as you have already mentioned other details (version, algorithm, etc.) in the data and method section

Line 6: measured "validly mostly in "? Please rewrite

Page 21: Line 7-9: If you have selected the measurements in accordance with the validation guidelines, then all measurements are valid and good for scientific analyses. Why do you want to write again about the validity?

satellite foot-prints and strips: the satellite measurements are on certain latitude/ longitude grids and you need to process the data accordingly. Therefore, this is an issue of data processing, not a problem of satellite measurements.

Page 34, Line  8: "we will make use of", and delete more

Page 34, l 26-27: Besides, soil moisture, workability and trafficability might improve the……. Please rephrase this sentence.

Page 35: Line 3-5:  What is more, because only modeled ammonia columns at overpass time are selected for averaging, …  Please rewrite it.

Page 35, 5-6: Dammers et al. (2016) found that the validity of the IASI product is quite limited because the satellite retrievals are biased. This statement is vague. Please provide details of the bias (e.g. region, period of comparison, compared measurements, etc).

**Page 36**

Line 2: The other reason could be the threshold of De Martonne-Index applied to the area of interest. Please give details of this.

Line 13: "Ideally" is enough

Line 14: "would be most optimal in this paper", is this topic discussed in this paper?

How much is (ammonia emissions) from the manure transport?

Line 26-27: "The distribution of annual emission obtained from the updated model is similar to that from the original MACC-III model". However, you have stated that the model has improved. I am confused, what is the improvement then?

Page 41: Why the indices are very high at the end of the year?

---

## Author Comment (AC2) · 27 Aug 2020

Thanks for your feedback on our paper.

We adjusted the structure of the paper. We moved the temporal allocation of emissions from grazing, animal housing and manure storage to the appendix in order to shorten the length. In addition, we went through the manuscript and rephrased the sentences to improve the style of writing.

The Dutch MAN network started measuring ammonia in 22 nature areas and then was expanded to 84 areas in 2019. The stations are located as far from emission sources

as they can get, but a few of them are still close to source since the Netherlands is small in size. As a result, the majority of the MAN stations could reflect ammonia level over a wider area.

---

## Author Comment (AC3) · 27 Aug 2020

Thank you for the time and effort you have devoted to offer valuable feedback on my manuscript.

We have gone through the manuscript and corrected the issues we found. We adjusted the structure of the paper. We moved the temporal allocation of emissions from grazing, animal housing and manure storage to appendix in order to shorten the length.

Multi-year model run can help to test seasonal cycle and inter-annual differences which are caused by inter-annual changes in meteorology and land use. Difference in land

use caused by crop rotation will affect manure and subsequent ammonia emission distribution. At the moment, INTEGRATOR land use data is only available for 2010. Moreover, meteorology has impact on ammonia emission estimates in two aspects. First, as described in this paper, temperature, precipitation and wind speed shape emission time profile. Second, the same factors also contribute to difference in emission fraction which linearly correlates emission and N applied. Therefore, after the ongoing work on crop mapping and emission fraction modeling, we will then look at multi-year model run.

Germany occupies the majority of grid cells in the area of interest. Therefore, when we described the spatial characteristics of ammonia emission, the changes in Germany were more visible due to its size. However, from the performance assessment by comparing annual averaged total columns in Table 2, one can see that the improvement in Luxembourg is the most significant, followed by that in the Netherlands. Table 3 indicates that the improvement is more apparent in the Netherlands by comparing surface concentrations.

---

## Author Response (AR1)

17 September, 2020

Dear Dr. Kuttippurath,

Please find enclosed a revised version of the manuscript entitled "Modeling Atmospheric Ammonia using Agricultural Emissions with Improved Spatial Variability and Temporal Dynamics" that I would like to resubmit to "Atmospheric Chemistry and Physics".

In this revised manuscript we have taken account of all the comments by interactive reviewers (see the replies to their comments).

In line with one of the reviewers, we redid the analysis by comparing our results with the complete daytime IASI total column dataset. Besides, in order to shorten the length of the main body of the paper, we updated methodological descriptions and moved some of them to appendix. In addition, we further included suggestions from the reviewers, namely to shorten sentences and rewrite them in a clear way. Furthermore, as the reviewers pointed out, we moved certain paragraphs in the result section to the discussion section. In this document, you will find the point-by-point response to the reviewers and the marked-up paper showing all the changes that were made. We also upload a clean version separately.

I look forward to hearing from you soon.

Kind regards,

Xinrui Ge (on behalf of all co-authors)

Wageningen University and Research Centre

Mail: P.O. Box 47, 6700 AA Wageningen, the Netherlands

Phone:          +31 317 486514

E-Mail:          xinrui.ge@wur.nl

**Point-by-point response to the reviews**

**Comments from Reviewer 1**

- **Comment 1**: *It might not be sensible trying to consider all issues in a single paper. It results in a very long paper but nevertheless limits the amount of detail that can be presented for each of the issues.*
  **Response**: This paper describes how we improved the spatial details and temporal characteristics (based on spatial details) of NH3 emission for LOTOS-EUROS to better predict NH3 concentrations in space and time that were subsequently compared with in-situ surface concentrations measurements and remotely sensed total columns from IASI for validation. We do not see how we can remove parts, since comparing the results with satellite and in-situ measurements, allows us to recognize needed improvements in the model and the measurements.

- **Comment 2**: *It is unclear how the livestock numbers per category per square kilometer are obtained. The same is true for the distribution of mineral nitrogen fertilizer. This is represented by the gap between Figure 1 and 2. Figure 1 gives a high-level overview of the modeling system whereas Figure 2 gives a quite detailed description of how excretion per livestock category per square kilometer is used to estimate ammonia emissions.*
  **Response**: Indeed, the downscaling of livestock was not described. We now added this by including the following text: *The data on livestock numbers in various animal categories at NUTS2 level have been downscaled to a 1 km2 resolution using expert-based judgment with spatial data sources on land use, slope, altitude and soil characteristics influencing the livestock carrying. A major distinction was made between grazing animals and other animals. Dairy cows, beef cattle, sheep and goats were assumed to be highly dependent on local land resources for grazing or feed production. Pigs and poultry were assumed to be held in more land independent systems. For more detailed information on the downscaling of livestock, we refer to Neumann et al. (2009)*

- **Comment 3**: *If the authors cannot identify a published report that describes the details of their INTEGRATOR model, it would be adequate (but less satisfactory) to put details in the Supplementary Material / In a number of cases there are references to De Vries et al. (2011) that this does not appear in the reference list.*
  **Response**: We now added the reference to De Vries et al. (2011). In addition, we gave more information on the ammonia emission fractions for housing and manure storage, grazing, manure application and fertilizer application, while referring to a recent report by de Vries et al (2020) as given below.
  - *De Vries, W., Leip, A., Reinds, G. J., Kros, J., Lesschen, J. P. and Bouwman, A. F.: Comparison of land nitrogen budgets for European agriculture by various modeling approaches, Environ. Pollut., 159(11), 3254–3268, doi:10.1016/j.envpol.2011.03.038, 2011.*
  - *De Vries, W., L Schulte-Uebbing, J. Kros and J.C. Voogd, 2020. "Assessment of spatially explicit actual, required and critical nitrogen inputs in EU-27 agriculture" Wageningen, the Netherlands, WenR rapport (in press)*

- **Comment 4**: *The authors are quite occupied with some of the finer-scale details of manure regulations (e.g. the application of manure on Sundays not being permitted) but as far as I can see, not such aspects as whether low-emission manure application techniques are mandatory.*
  **Response**: This is accounted for in view of housing and manure storage in INTEGRATOR. Emission fractions for NH3 emissions from housing and manure storage are distinguished per animal type

(6 categories) and manure type (liquid vs. solid for 3 animal categories). For some countries, the basic emission fractions are modified because assumptions on implementation of low-emission manure storage or housing systems. For these countries, a new emission fraction is calculated based on the degree of implementation of emission-reducing technologies, and the reduction efficiency of the technology.

- **Comment 5**: *The authors did not choose to compare their simulations with measurement sites that did not have local ammonia sources, as such sites would be expected to reflect the consequences of agricultural practices over a wider area.*
  **Response**: We mention in the paper that the setup of monitoring sites is such that measurement data should be representative of a wider region. The spatial resolution of the updated model is around 7km by 7km, there may always be some impacts of local sources. Most ideally, a station next to arable land but is distant from an animal house or manure storage would be most optimal to verify the timing of emission from manure/fertilizer application obtained with the methodology of the TIMELINES model. However, we did not have the information to select sites to avoid those influences. We thus used all sites and mentioned that local influences cannot be removed completely.

- **Comment 6**: *The manuscript requires some attention regarding language and typographic errors.*
  **Response**: Thanks for pointing out this issue. I have done another review to make corrections.

**Comments from Reviewer 2**

- **Comment 1**: *The paper focuses on modeling improvements of agricultural emissions. This study is well written and is definitely valuable for the atmospheric modeling communities. I would thus recommend submitting the paper in a more appropriate journal (such as Geoscientific Model Development).*
  **Response**: We agree that the paper is valuable for the atmospheric modeling communities but also for the atmospheric chemistry and physics community and we thus like to have the paper in ACP.

**Comments from Reviewer 3**

- **Comment 1**: *It is recommended for public discussion.*
  **Response**: Thank you for your positive feedback.

**Comments from Matthieu Pommier (27 Apr 2020)**

- **Comment 1:** *In the use of the positive data, I also suggest using the negative columns. … On the last filter, the authors should justify the 75% threshold. The larger errors given in the IASI data set are often related to the lower NH3 total columns and, as with the filter on positive columns, the authors take the risk to overestimate their IASI averaged distribution. … Thus, I suggest using the full data set.*

  **Response:** During the comparison of annual averaged columns, we filtered the measurements based on relative error (and other criteria) to ensure that we used observations with smaller uncertainties. We think you have a point that if we do so, the averaged columns will be most likely overestimated because smaller columns tend to have larger relative errors and to be excluded.

As a matter of fact, filtering based on either relative error or absolute error will lead to biases in the outcomes. As what Dammers et al. (2017) pointed out, we then used all measurements.

Therefore, we adopted your suggestion and used all measurements (including negative columns) for validation. The background level in annual averaged IASI columns has been eliminated, the comparison was subsequently conducted again, and the results were updated accordingly in the paper. In conclusion, the use of all IASI data will greatly help to eliminate the background level of ammonia and improve the linear correlation between observed and model columns, but it does not have a large impact on what has been discussed in the paper. The plots here will be updated in the final version of the paper. Thanks for your time!

**Comments from Anonymous Referee #1 (24 Jun 2020)**

- **Comment 1:** *This remains a very long paper and some parts resemble more a good working draft written by a PhD student than a final draft that has had the guiding hand of an experienced scientist. In this respect, the authors are doing themselves a disfavour, since the work is otherwise something of a tour de force. In particular: - The English needs to be improved.*

  **Response:** We adjusted the structure of the paper. We moved the temporal allocation of emissions from grazing, animal housing and manure storage to appendix in order to shorten the length. In addition, we went through the manuscript and rephrased the sentences to improve the style of writing.

- **Comment 2:** *As I noted in my original review, I am surprised that the authors did not choose to compare their simulations with measurement sites that did not have local ammonia sources. Such sites would be expected to reflect the consequences of agricultural practices over a wider area*

  **Response:** The Dutch MAN network started measuring ammonia in 22 nature areas and then was expanded to 84 areas in 2019. The stations are located as far from emission sources as they can get, but a few of them are still close to source since the Netherlands is small in size. As a result, the majority of the MAN stations could reflect ammonia level over a wider area.

**Comments from Jayanarayanan Kuttippurath (20 Jul 2020)**

- **Comment 1:** *The MS is too long and was also a bit difficult to read because of the style of writing. Therefore, kindly consider shortening the length of the article and rephrase the sentences as suggested below, to the least. I have given only some examples. Please go through the entire article and check.*

  **Response:** We adjusted the structure of the paper. We moved the temporal allocation of emissions from grazing, animal housing and manure storage to appendix in order to shorten the length. Thank you for the effort to point out the examples of issues in the style of writing in the manuscript. We have gone through the manuscript and corrected the issues we found.

- **Comment 2:** *I thought one more year is needed to test the new model so that seasonal cycle can be tested and inter-annual differences can examined. If you have the additional model runs, please include and discuss.*

**Response:** Multi-year model run can help to test seasonal cycle and inter-annual differences which are caused by inter-annual changes in meteorology and land use. Difference in land use caused by crop rotation will affect manure and subsequent ammonia emission distribution. At the moment, INTEGRATOR land use data is only available for 2010. Moreover, meteorology has impact on ammonia emission estimates in two aspects. First, as described in this paper, temperature, precipitation and wind speed shape emission time profile. Second, the same factors also contribute to difference in emission fraction which linearly correlates emission and N applied. Therefore, after the ongoing work on crop mapping and emission fraction modeling, we will then look at multi-year model run.

- **Comment 3:** *I find the improvement (the difference too) is mostly for the simulations for Germany, not for other countries. Is there any reason for this?*

**Response:** Germany occupies the majority of grid cells in the area of interest. Therefore, when we described the spatial characteristics of ammonia emission, the changes in Germany were more visible due to its size. However, from the performance assessment by comparing annual averaged total columns in Table 2, one can see that the improvement in Luxemburg is the most significant, followed by that in the Netherlands. Table 3 indicates that the improvement is more apparent in the Netherlands by comparing surface concentrations.

**List of all relevant changes**

1. We added more references to describe the details of their INTEGRATOR model: De Vries et al., 2011 and De Vries et al., 2020.
2. The method to allocate manure according to crop type was moved to Appendix B. Even though it is an essential process emission estimates, we focus more on the spatial and temporal distribution.
3. Temporal allocation of ammonia emission from grazing, animal housing and manure storage was moved to Appendix D to reduce the length of the paper. Because compared to the emission time profiles of manure and fertilizer application, the modifications applied to the original equations are relatively less.
4. The paragraph discussing the timing of peak emission after manure and fertilizer application was deleted. We replaced it by saying "We assumed that the peak of emission after application occurs at noon on the second day after the estimated central fertilization day."
5. We only kept the equation to derive the weekly De Martonne-Index.
6. We completed "appl." In Figure 2 as "application".
7. We enlarged the scale of the axes in Figure 3,7,8,9 so that they are more readable.
8. The two examples of time profile during development were moved to Appendix H.
9. When comparing annual averaged total columns between IASI measurements and modeled results, we used all daytime measurements without filtering based on relative or absolute errors. Therefore, Figure J1 in Appendix J illustrating the number of valid measurements was updated. As is shown in Figure 4(b), the background level of ammonia has been eliminated. Figure 6 shows much better linear correlation between the measured and modeled total columns. Table 2 demonstrates that the updated model performs even better after the inclusion of all daytime measurements. However, it still overestimates emission in lower latitude and underestimates in higher latitude.
10. We moved the paragraph after Table 2 to the discussion section.
11. We moved the paragraph before Figure 8 to the discussion section.
12. We modified the discussion section by re-organizing the original five subsections into two. We discussed what we found out by comparing surface concentrations and total columns, to offer a clear view to the readers.
13. We went through the paper and correct the typos. We also rewrote sentences that were difficult to read.

[revised manuscript text omitted]
25   When illustrating the comparison of concentrations time series, we selected several stations that are not close to local agricultural sources (as shown in Table 1 in Appendix I) so that the local influences on measurements could be minimized. Besides,  by comparing all individual measurements at all available -stations, the overall performance of the updated model can be determined.
30

**2.5.2 Satellite Observations**

 Infrared Atmospheric Sounding Interferometer (IASI)

5

 is a Fourier transform infrared (FTIR) spectrometer that measures the thermal infrared (TIR) radiation emitted by the Earth's surface and the atmosphere. It circles in a polar Sun-synchronous orbit and operates in nadir mode. It has a wide swath width of 2 x 1100 km, which corresponds to 2x15 mirror positions, while the spatial resolution is 50 km x 50 km, composed of 2 x 2 circular pixels. Each circular pixel is a 12 km diameter footprint on the

10 ground at nadir (Clerbaux et al., 2009).

An  improved $NH_3$ retrieval scheme for IASI spectra was presented by Van Damme et al. (2014), which relies on the calculation of a dimensionless Hyperspectral Range Index (HRI). Whitburn et al. (2016) continued with HRI and introduced a neural-network-based algorithm to obtain $NH_3$ total columns. Van Damme et al. (2017) made some improvements by training separate neural networks for land and sea observations,

15 enhanc thermal contrast, and introducing a bias correction over land and sea and the treatment of  satellite zenith angle, which resulted in the latest product Artificial Neural Network for IASI ANNI-NH3-v2.1. As is pointed out by Van Damme et al. (2017), weighted averaging is no longer recommended in ANNI-NH3-v2.1, arithmetic mean or median is suggested if averaging has to be performed.

Regardless of the improvement of $NH_3$ column retrieval from satellite observations, there is still substantial variability in

20 measurement uncertainty, varying from 5% to over 1000 % (Van Damme et al., 2017). Measurements with small magnitude tend to have larger relative uncertainties. Due to considerable uncertainties and the requirement of clear-sky conditions, IASI data is insufficient for real-time monitoring but sufficient if used to calculate monthly or yearly average distributions. In this study, the annual mean was compared with LOTOS-EUROS output for verification. The monthly mean was calculated to investigate the feasibility of being used for validation of temporal variability. For each IASI observations, the modeled results

25 that are closest in space and time were selected.

In this paper, we used  ANNI-NH3-v2.2R-I IASI dataset which was obtained with  ECMWF ERA-Interim meteorological data and surface temperature data retrieved from a dedicated network. After  the dataset was downloaded from the AERIS portal (https://iasi.aeris-data.fr/NH3R-I_IASI_A_data/)

30 selected satellite observations with daytime overpass because daytime is the better time to measure $NH_3$ (Van Damme et al., 2017). Area-weighted annual mean was derived by resampling  the circular footprints of IASI onto the grid used in LOTOS-EUROS. Area

avering  also applied to the calculation of mean relative error of each grid cell. Finally, post-filtering was carried out to obtain more reliable distributions: all grid cells with less than ten measurements  were rejected.

5  ~~Regardless of the improvement of $NH_3$ column retrieval from satellite observations, there is still a very large variability in measurement uncertainty, varying from 5% to over 1000 %. One major source of measurement uncertainties comes from the thermal contrast retrieval algorithm which results in the variable sensitivity of the outgoing infrared radiation to the lower troposphere (Clarisse et al., 2010; Van Damme et al., 2017). An accurate temperature profile is needed, without which larger measurement errors will occur. Summer daytime is the best time to measure $NH_3$ while winter nighttime is the worst (Van~~

10
~~, with summer daytime being the best time to measure $NH_3$ while winter nighttime being the worst (Van Damme et al., 2017). Because thermal contrast, whose accuracy needs to be improved, leads to the variable sensitivity of the outgoing infrared radiation to the lower troposphere (Clarisse et al., 2010; Van Damme et al., 2017). As a result, we only use satellite observations with daytime overpass in 2010.~~

15  ~~Due to large uncertainties in the measurements and requirement of clear-sky conditions, the number of valid 
[revised manuscript text omitted]

[Figure]

Averaged Relative Error of IASI Measurements in 2010 (a); Annual Averaged IASI $NH_3$ Total Columns in 2010 (b)

Figure 4 The map of area-averaged relative error of IASI daytime measurements in 2010 (a). The map of area-averaged total columns after filtering out grid cells with less than ten valid measurements and an averaged relative error larger than 75% (b).

5   The modeled annual averaged total columns from LOTOS-EUROS simulations are shown in Figure 5Figure 6. , which are calculated from 3-dimensional concentration outputs that are spatially and temporally closest to the satellite measurements overpass. Overall, the updated result (Figure 5(b)) obtained with the updated annual emission distribution and time profiles in Figure 6(b) gives a higher magnitude of $NH_3$ columns than the original one. Large relative differences that are more than 100% occur mostly over Germany and the Eastern Netherlands. The hot spots in the original simulations in the Eastern Netherlands,

10   Nordrhein-Westfalen and Niedersachsen in the original simulations expand prominently to a much more extensive domain standing outin the new simulation. Moreover, new hot spots are witnessed in other regions in Germany, such as Bayern and Baden-Württemberg close to the border with Austria and Switzerland.

[Figure]

15   **Figure 5 Simulated annual averaged total columns from LOTOS-EUROS using the original MACC-III annual emission distribution and static time profile (a) and using MACC-INTEGRATOR emission totals and updated time profiles (b).**

Figure 6Figure 7 shows scatter plots comparing IASI observations and LOTOS-EUROS column estimates, with the left and right panel comparing the measurements with the original modeled result and the right panel comparing the measurements with the updated output, respectively. Figure 6Figure 7(a) and Figure 6Figure 7(b) include all grid cells in Germany and Benelux. While simulated total columns could reach towards 0 $molec/cm^2$, IASI measurement has a minimum value of

25   approximately $0.8 \times 10^{16} molec/cm^2$, which validates our observation from Figure 5(b). The simulated total columns from the original model are mostly underestimated. , Meanwhile, there exist both overestimation and underestimation in the updated output. One can see in Fig. 7(b) that there are tTwo plumesclusters appear in Figure 6(b), with 
[revised manuscript text omitted]